# The influence of garden spatial configuration on tourist behavior: A systematic review based on Space Syntax

**Bin Li** [1], **Mohammad Mujaheed Hassan**[1]*, **Yan Han**[2], **Jasmine Leby Lau**[1]

1 Faculty of Human Ecology, Universiti Putra Malaysia, Serdang, Selangor, Malaysia, 2 Department of Spatial Culture Design, Graduate School of Techno-Design, Kookmin University, Seoul, Korea

☯ These authors contributed equally to this work.
* mujaheed@upm.edu.my

## Abstract

As composite spaces that integrate nature and culture, gardens are no longer regarded as merely static objects of visual appreciation in the context of urbanization, but have become essential venues for public cultural tourism and leisure. Consequently, the behavioral characteristics of tourists in gardens have attracted increasing academic attention. Space syntax, as a tool for analyzing the influence of spatial organization on human behavior, quantifies spatial configuration characteristics and can reveal how garden spatial configuration affects tourists' movement paths and spatial preferences, thereby enabling a systematic examination of the impact of space syntax–based garden spatial configuration on tourist behavior. adheres to the Following by PRISMA 2020 guidelines, this study conducted a literature search for the period 2015−2015 in four databases, namely Web of Science, Scopus, JSTOR, and ScienceDirect Based on explicit inclusion and exclusion criteria, 16 high-quality empirical studies were ultimately selected. Results indicate that indicators such as integration, connectivity, and depth, demonstrate significant explanatory in predicting tourist path selection, stay locations, and spatial preferences. Furthermore, the influence of spatial structure on visitor behavior is not a singular direct effect. Visitor perceptions, particularly aesthetic preferences, cultural cognition, and sense of security, play a crucial mediating role between spatial structure and behavior. Based on these findings, this study proposes the "Structure–Perception–Behavior (SPB)" framework. Its cross-scale methodological insights provide a theoretical foundation and practical pathway for subsequent landscape space optimization design and visitor behavior guidance.

## 1. Introduction

As an art form, Gardens integrate artistic elements such as plants, water features, topography, architecture, and ornamental structures to create significant spatial

**Data availability statement:** All relevant data are within the manuscript and its Supporting information files.

**Funding:** The author(s) received no specific funding for this work.

**Competing interests:** NO.

environments that combine cultural aesthetics with practical functionality [1]. Regarded as a "second nature," they fulfill both physiological and psychological human needs [2]. Across different civilizational lineages, the evolutionary trajectories and spatial connotations of garden types differ markedly. For instance, Eastern gardens, influenced by Confucian, Taoist, and Buddhist philosophies, emphasize the creation of spaces that embody the "unity of heaven and humanity," personal cultivation, and transcendent mental states [3]. Within this framework, Chinese imperial gardens, shaped by ritual systems and imperial discourse, emphasize grand layouts and central axis order, symbolizing political power [4]. Private gardens, however, favored the "microcosmic" landscape aesthetic, emphasizing the literati's appreciation of shifting vistas and self-cultivation through "scenes that change with every step [5]."Japanese gardens, inheriting early Chinese Buddhist traditions, ultimately evolved under the influence of Zen and the tea ceremony to use stones, sand, and moss as primary elements, creating wabi-sabi aesthetics and meditative spaces [6]. In contrast to the Eastern pursuit of natural beauty, Western gardens emphasize the unity of religion and power through converging axes and waterways [7]. Examples include Renaissance gardens that express "rational domination over nature" through geometric order [8], and Baroque gardens that reinforce monarchical authority through spatial hierarchy [9]. Within contemporary urban contexts, gardens, whether rooted in Eastern traditions or Western lineages, have become spatial vessels for recreation, sightseeing, and social interaction for both residents and visitors.

People regard urban space as a green environment created in accordance with the laws of nature [10], the evolution of human demand for green spaces, from singular to diverse and from simple to complex, has promoted the development of urban gardens [7]. Consequently, visitor behavior within garden spaces has increasingly drawn interdisciplinary attention from fields such as urban planning, landscape architecture, tourism geography, and environmental psychology [11], the research focus on garden spaces has gradually shifted from cultural aesthetics to the influence of spatial design on visitor behavior [12]. Spatial syntax, proposed jointly by Bill Hillier and Julienne Hanson [13], primarily analyzes the relationship between urban spatial structures and human behavior. By integrating core metrics such as such as integration, connectivity, and depth, it reveals how spatial layouts influence the range of human activity [14]. For instance, it can uncover individual or group clustering patterns, path preferences, and spatial perceptions [15]. In recent years, researchers have increasingly applied spatial syntax methods to analyze complex garden spaces, this approach illuminates how intricate garden layouts shape clusters of visitor behavior, movement paths, and dwell-time hotspots, thereby filling methodological gaps in traditional qualitative studies [16].

Existing research consistently indicates that the spatial configuration of gardens directly influences visitors' behavioral choices and satisfaction levels. For instance, when garden structures, water features, and rockeries are obscured by towering vegetation, it impedes visitors' visual access, thereby diminishing their spatial perception [17]. Conversely, overly dense clusters of winding path junctions can induce spatial cognitive difficulties. Low integration and connectivity can lead to disorientation and

path uncertainty, thereby visitors' desire for spatial exploration and behavioral motivation [18]. Finally design that empha-size path meandering, while satisfying aesthetic intentions, can also pose challenges to directional recognition and cause spatial distress [19]. Lee et al. [20] found that installing recreational facilities along spatial edge without visual signage, still makes it challenging to attract visitors to use them. Furthermore, the presence of stairs and narrow passages in highly connected areas limits accessibility for older people and children, creating a sense of behavioral separation between these groups and others [21].

In summary, the spatial configuration of gardens exerts a significant influence on tourist behavior [22]. The necessity of this systematic review lies in the current lack of a comprehensive analysis employing Space Syntax to examine the relationship between garden spatial structures and visitor behavior. Although prior studies have confirmed the correlation between spatial configuration and tourist behavior, a lack of systematic reviews persists—particularly those integrating the explanatory power and adaptability of different spatial variables. Therefore, this study conducts a systematic review of selected literature employing Space Syntax-based approaches to examine the relationship between garden spaces and tourist behavior, and develops the analysis around the following key questions (Fig 1):

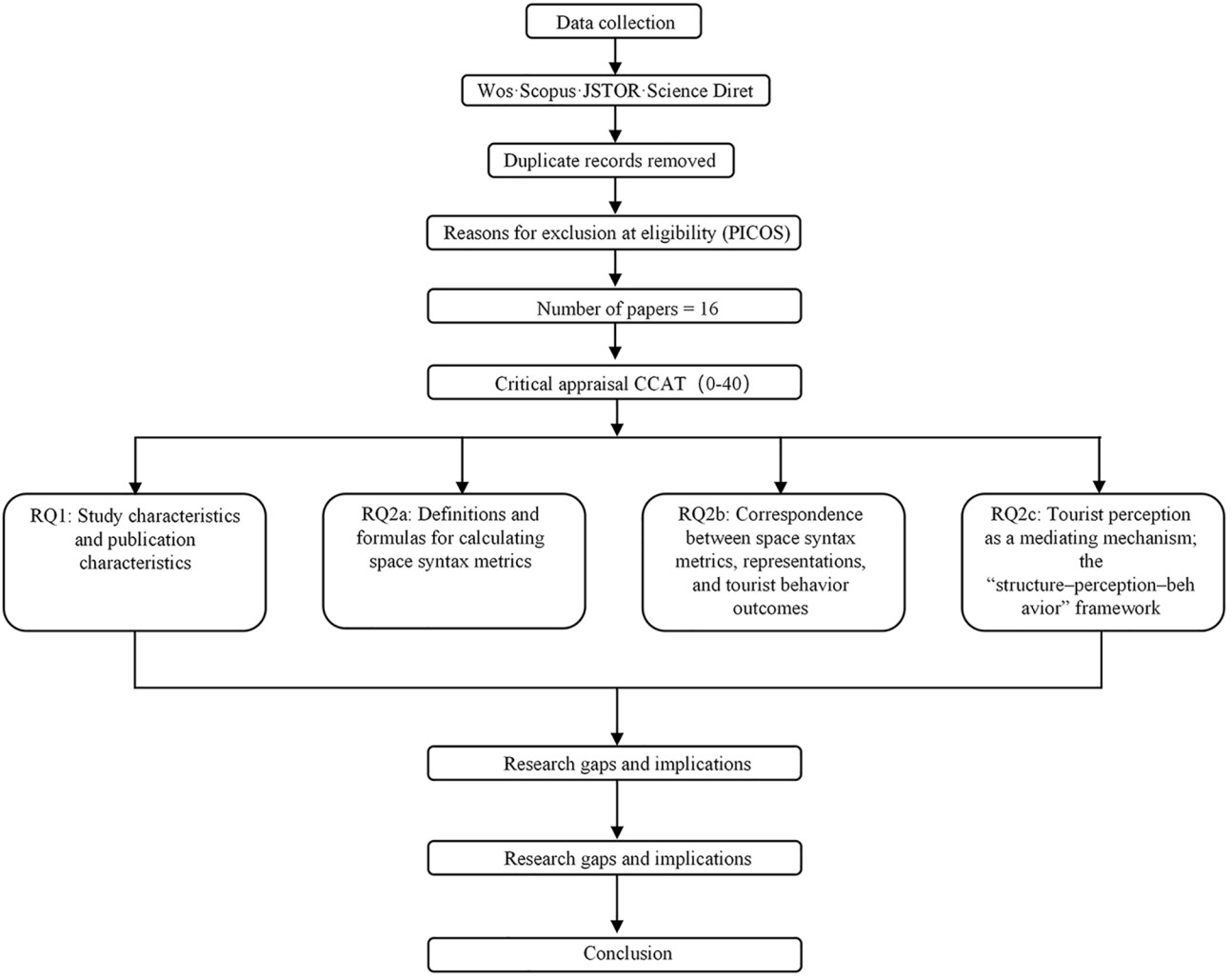

**Fig 1. Flow diagram for literature review.**

- RQ1: From 2015 to 2025, what distribution and evolutionary patterns are exhibited in the research characteristics and publication features of space-syntax-based studies on "garden spatial configuration and tourist behavior''?

- RQ2a: Which core metrics of space syntax were employed in the included studies? What are the definitions and computational formulas of these metrics?

- RQ2b: How do different metric characteristics and spatial representations in space syntax (VGA, Segment, Isovist, Convex) influence tourist behavioral outcomes?

- RQ2c: Which features have existing studies used to reveal the mediating role of tourist perception in the relationship between spatial configuration and behavioral outcomes?

## 2. Methods

This study follows the guidelines of the systematic review PRISMA (2020) [23], a title that has been registered on the international platform for registered systematic Evaluation of Meta-Analysis Programs under the registration number: INPLASY202560013.

### 2.1 Search strategy

In this study, electronic databases such as Web Of Science, Scopus, JSTOR, and ScienceDirect were systematically searched with a search deadline of January 2, 2025. The search process was matched with keywords by Boolean operators AND and OR (Table 1), and coordinated search terms and search strings were used uniformly for each database to ensure that all databases were searched consistently (S1 Table).

### 2.2 Criteria and quality assessment

This study employed the PICOS framework for literature screening [24], the inclusion criteria simultaneously satisfied the following conditions (Table 2). Although the search covered major databases, the number of studies ultimately included was relatively small; therefore, this study does not rely solely on statistical frequencies but adopts an interpretive synthesis, emphasizing the correspondence between "metrics–representations–behavior'' and the elucidation of the "structure–perception–behavior'' mechanism. The quality of the included literature was assessed using the Crowe Critical Appraisal Tool (CCAT) to enhance the precision of the research [25]. Developed by Lynne Crowe, the tool is applicable to quantitative, qualitative, and mixed-methods studies and provides a standardized evaluation framework [26].The assessment

**Table 1. Search string.**

| Search Builder | Search String |
| --- | --- |
| Space Syntax | "Space" AND "syntax" OR "spatial" AND "syntax" |
| Garden | "garden" OR "park" OR "grove" |

**Table 2. PICOS criteria for inclusion of studies.**

| Items | Detailed inclusion criteria |
| --- | --- |
| Population | Tourist engagement in garden, park, and other landscape environments |
| Intervention | Spatial structural characteristics of gardens |
| Comparison | Without a control group |
| Outcome | Involving tourist behavioral performance |
| Study design | Empirical studies using space syntax analysis |

comprises the following dimensions: introduction, background, methods, abstract, data collection, ethics, results, and discussion. Each dimension is scored from 1 to 5, with no half points [27](S2 Table).

## 2.3 Study selection

In accordance with PRISMA 2020, 1,040 records were retrieved from four databases; after removing 119 duplicates, 921 proceeded to title/abstract screening: 172 were excluded for timeframe mismatch, 65 were books, and 488 were unrelated to the topic. A total of 200 full texts were obtained and assessed: based on PICOS, 59 reviews/non-empirical studies, 11 without space syntax, 9 without tourists/visitors, 91 outside garden/park/woodland contexts, and 8 abstract-only/no full text were excluded, 5 theses, 1 pilot preprint. Ultimately, 16 studies were included (Fig 2). In cases of disagreement during the screening process, a third expert was consulted to assist in reaching a final consensus.

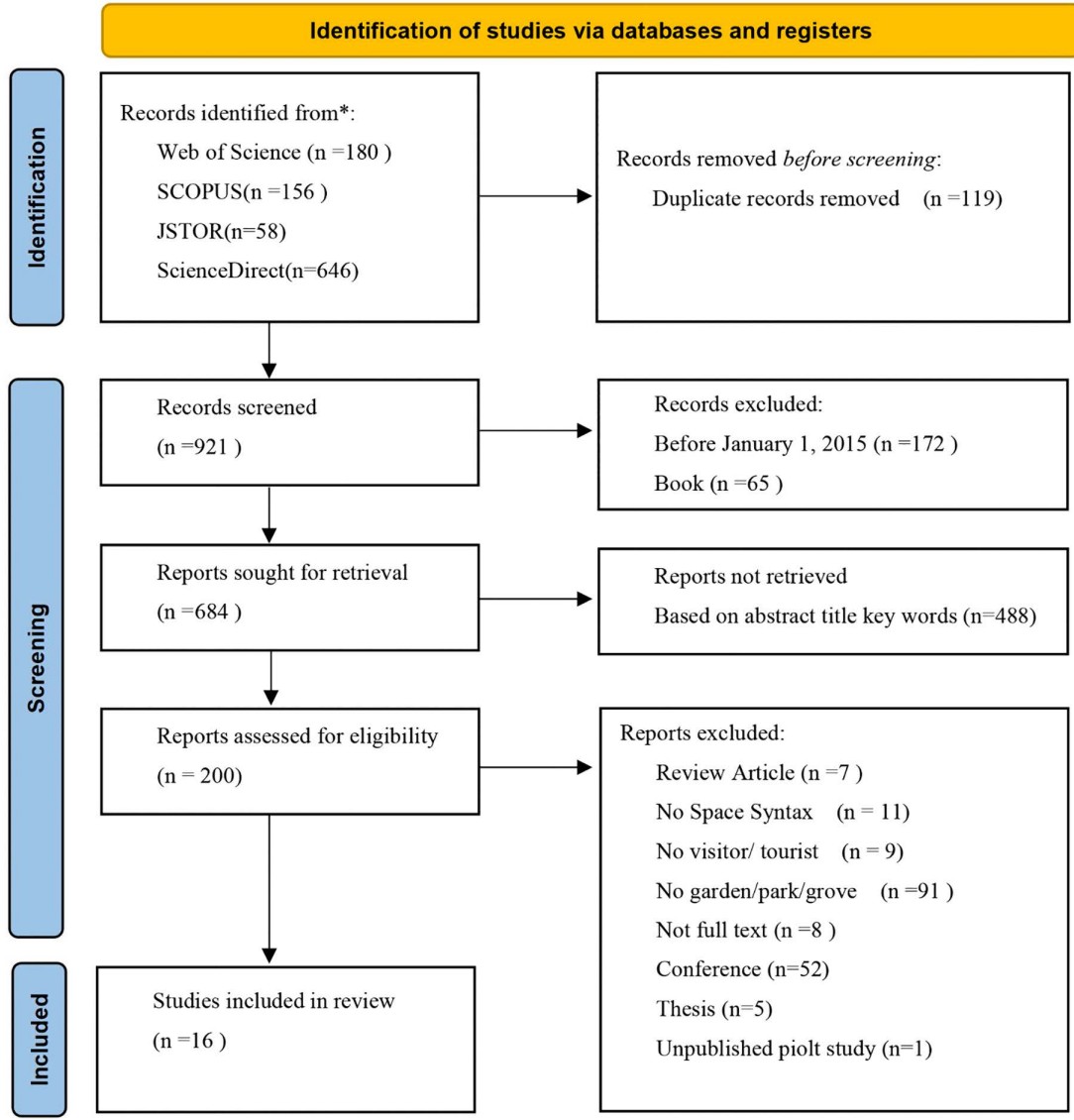

**Fig 2. PRISMA flow diagram.**

## 3. Results

A total of 1,040 studies were screened for this review, with the specific reasons for exclusion detailed in the PRISMA flow diagram (Fig 2). Ultimately, 16 studies met the inclusion criteria and were evaluated using the Crowe Critical Appraisal Tool (CCAT). The distribution of quality scores is presented in Table 3. The results of the quality appraisal indicate that all included studies demonstrated a high level of overall quality.

### 3.1 Research characteristics

This study provides a systematic synthesis of the 16 included studies (Table 4). The general characteristics were summarized across the following dimensions: country of publication, study location, spatial analysis, and space syntax modeling methods, core space syntax metrics (integration, connectivity, choice, control), radius or weighting settings, and reported outcomes. Specifically, spatial analysis and space syntax modeling methods comprised two categories: (1) visualization- or statistics-based analyses, including kernel density estimation (KDE), heatmaps, and experiential maps; and (2) space syntax modeling methods, including axial maps, visibility graph analysis (VGA), segment analysis, and isovist-based models. Notably, axial, segment, and VGA analyses emphasize the global network structure or connectivity, whereas the isovist analysis focuses on local field-of-view visibility.

### 3.2 Literature sources and publication trends

This study included 16 representative publications spanning 2015–2025. The research regions encompassed China, Poland, Pakistan, and South Korea (Fig 3). Notably, over the past decade, Chinese and South Korean scholars have produced the highest volume of publications exploring garden spaces and visitor behavior. It is worth noting that although Malaysia, Egypt, Turkey, Iran and Pakistan have produced fewer publications, these studies also offer valuable complementary perspectives for researchers in other fields.

**Table 3. Quality of studies assessed using the crowe critical appraisal tool (CCAT).**

| Study | P | I | De | S | Dc | EM | R | Di | T |
|---|---|---|---|---|---|---|---|---|---|
| Zhai et al. (2018) [22] | 5 | 5 | 4 | 4 | 5 | 4 | 4 | 5 | 36 |
| Huang and Lee (2023) [28] | 5 | 4 | 5 | 4 | 5 | 3 | 5 | 4 | 35 |
| Zhang et al. (2020) [29] | 5 | 5 | 5 | 4 | 5 | 3 | 5 | 5 | 37 |
| Wu et al. (2025) [30] | 5 | 5 | 5 | 4 | 5 | 4 | 5 | 5 | 38 |
| Chen and Yang (2023) [31] | 4 | 5 | 5 | 5 | 4 | 3 | 4 | 5 | 35 |
| Yu et al. (2016) [32] | 4 | 4 | 5 | 3 | 5 | 3 | 4 | 4 | 32 |
| Gomaa et al. (2024) [33] | 5 | 5 | 5 | 4 | 5 | 4 | 5 | 5 | 38 |
| Lee (2021) [34] | 4 | 4 | 5 | 3 | 5 | 3 | 4 | 4 | 32 |
| Saadativaghar & Zarghami (2023) [35] | 5 | 5 | 5 | 4 | 5 | 4 | 5 | 5 | 38 |
| Chen & Yang (2023) [36] | 4 | 4 | 5 | 4 | 5 | 3 | 4 | 4 | 33 |
| Mohammadi & Ujang (2022) [37] | 4 | 4 | 4 | 3 | 5 | 3 | 4 | 4 | 31 |
| Yu et al. (2021) [38] | 5 | 4 | 5 | 4 | 5 | 3 | 5 | 4 | 35 |
| Mohamed et al. (2023) [39] | 5 | 5 | 5 | 4 | 5 | 4 | 5 | 5 | 38 |
| Zhang et al. (2019) [40] | 4 | 4 | 4 | 3 | 4 | 3 | 4 | 4 | 30 |
| Traunmüller et al. (2023) [41] | 5 | 5 | 5 | 4 | 5 | 4 | 5 | 5 | 38 |
| Chen et al. (2025) [42] | 5 | 5 | 5 | 4 | 5 | 4 | 5 | 5 | 38 |

NOTE: P, Preliminaries; I, Introduction; De, Design; S, Sampling; Dc, Data Collection; EM, Ethical Matters; R, Results; Di, Discussion; T, Total.

**Table 4. Characteristics of the studies.**

| Authors (Year) | Study sites | Spatial Analysis and Modeling Methods | Spatial Syntax Metrics | Radius/Weight Settings | Outcome |
|---|---|---|---|---|---|
| Zhai et al. (2018) China [22] | Urban forest park | Modified convex map (stroke-based) | Integration, Control, Connectivity | Global (Rn), metric weighting | Accessibility and tourist path behavior |
| Huang & Lee (2023) [28] Korea | Hefei urban park | Kernel density estimation (KDE) + heatmap | Integration | Global and local | Accessibility and space use |
| Zhang et al. (2020) [29] China | Lion Grove Garden | VGA | Visibility Graph, Integration | – | Behavioral patterns (route choice) |
| Wu et al. (2025) [30] China | The Three Gardens of Yangzhou | VGA+Segment | Integration, Connectivity, Choice | Local (R3/R5) + Global (Rn), angular weighting | Differences in spatial configuration and tourist perception |
| Chen & Yang (2023) [31] China | Humble Administrator's Garden | VGA | Visibility Graph, Isovist | – | Tourist perception and experience |
| Yu et al. (2016) [32] China | Suzhou· Yuyuan Garden | Segment+ VGA | Global Integration, connectivity, Control | – | Accessibility and staying behavior |
| Gomaa et al. (2024) [33] Pakistani | Peshawar Park | Axial | Integration, Step depth, Choice, Connectivity | Global+step depth from the main entrance | Accessibility and perception |
| Lee (2021) [34] Korea | Cheonan urban park | Segment | Integration, Visual connectivity | Global and local | Accessibility and space use |
| Saadativaghar & Zarghami (2023) [35] Iran | Eram Park, Hamadan, Iran | Axial | Connectivity, Integration, Depth, Control, Line length, Intelligibility | Local and global | Psychological restoration |
| Chen & Yang (2023) [36] China | Humble Administrator's Garden | Isovist | Integration, Depth, Visual area | Path-based, mean depth | Tourist experience and route choice |
| Mohammadi & Ujang(2021) [37] Malaysia | Kuala Lumpur urban park | Experiential maps | Local Integratio, Visual accessibility | – | Social interaction and accessibility |
| Yu et al. (2021) [38] China | Ningbo Tianyi Pavilion Museum Garden | convex | Integration, Choice, Width, Length, Enclosure ratio, Seating | – | Distribution of staying and spatial attributes |
| Mohamed et al. (2023) [39] Egypt | New Damietta urban park, Egypt | Visibility Graph | Integration, Connectivity, Choice | Global and local | Accessibility and tourist experience |
| Zhang et al. (2019) [40] China | Lion Grove Garden | VGA | Visual control, Revisiting proportion, Speed | Local and global | Sightline design and tourist distribution |
| Traunmüller et al. (2023) [41] Turkey | 42 community parks in Izmir | Segment+Axial | Integration, Choice, Connectivity | Multiple radii (R100–R2000, Rn), angular weighting | Park use intensity and accessibility differences |
| Chen et al. (2025) [42] China | Xiao Canglang Water Courtyard | Isovist | Isovist Area | Based on eight viewpoints and different visiting routes | Tourist visual experience and spatiotemporal perception |

NOTE: VGA (Visibility Graph Analysis): visibility graph maps used to represent visual fields in space. Segment: line segment model, which can be weighted by angle, length, or metric distance. Axial: axial map representing the longest and fewest sight lines. Isovist: the visible space from a given point under conditions of spatial occlusion. Experiential maps: records of participants' subjective experiences and behaviors within the site. Rn: global radius. Rk: local radius based on topological step depth (R3 or R5).

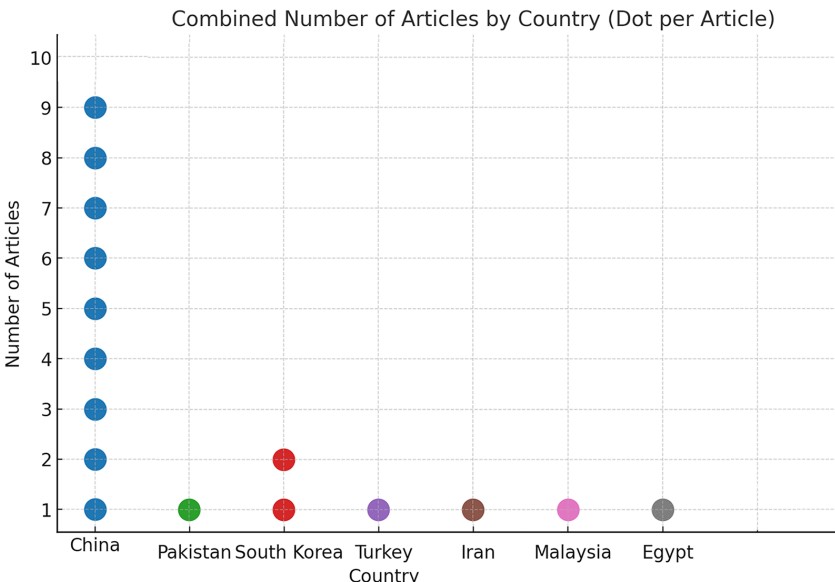

**Fig 3. Geographical distribution by Country.**

Fig 4 clearly shows the annual publication trend in this field. Results indicate that over the past five years (2015–2021), publication volume remained low, with instances of zero publications occurring, reflecting that this field has not garnered significant attention or favor among scholars. Since 2021, publication volume has surged dramatically, peaking in 2023 (N = 6), indicating the expanding application of spatial syntax methods within landscape architecture.

Research on the relationship between landscape spaces and visitor behavior has been primarily published in the following journals (Table 5). The top three journals are Urban Forestry and Urban Greening, Landscape Research, Asian Journal of Architecture and Construction Engineering, and Sustainability, each having published two papers on this topic over the past decade. The remaining journals published only one article related to this theme. During the literature search, this study found that publications appeared not only across core journals in fields such as architectural planning and design and landscape architecture, but also across interdisciplinary journals. This reflects the growing demand for cross-disciplinary integration within the academic community.

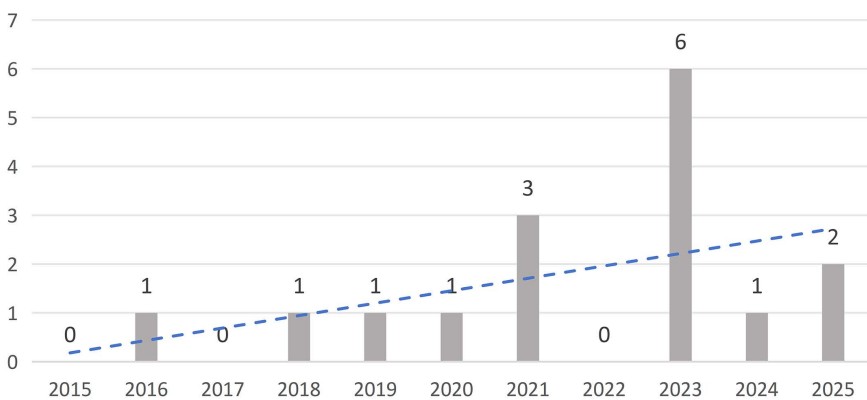

**Fig 4. Publication trends from 2015 to 2025.**

**Table 5. Journals and year of publication distribution.**

| Journal | 2015 | 2016 | 2017 | 2018 | 2019 | 2020 | 2021 | 2022 | 2023 | 2024 | 2025 | Totals |
|---|---|---|---|---|---|---|---|---|---|---|---|---|
| Urban Forestry & Urban Greening | | – | – | 1 | – | – | – | – | 1 | – | – | 2 |
| Frontiers of Architectural Research | | – | – | – | – | – | – | – | | – | 1 | 1 |
| Journal of the Korea Institute of Spatial Design | | – | – | – | – | – | – | – | 1 | – | – | 1 |
| Landscape Research | | – | – | – | 1 | – | – | – | 1 | – | – | 2 |
| Journal of Asian Architecture and Building Engineering | | – | – | – | – | – | – | – | 1 | 1 | – | 2 |
| Journal of Asian Architecture and Building Engineering | | – | – | – | – | – | – | – | – | – | 1 | 1 |
| Sustainability | | – | – | 1 | – | – | – | – | 1 | – | – | 2 |
| Visualization in Engineering | 1 | – | – | – | – | – | – | – | – | – | – | 1 |
| Civil Engineering and Architecture | | – | – | – | – | – | – | – | – | 1 | – | 1 |
| Archnet-IJAR: International Journal of Architectural Research | | | | | | | 1 | | | | | 1 |
| Urban Science | | | | | | | 1 | | | | | 1 |
| Journal of the Korea Institute of Spatial Design | | – | – | – | – | – | 1 | – | – | – | – | 1 |
| **Total** | 0 | 1 | 0 | 1 | 1 | 1 | 3 | 0 | 5 | 2 | 2 | 16 |

The findings from Sections 3.1 and 3.2 addresses RQ1: "What are the main trends in research based on Space Syntax exploring the relationship between garden spatial configuration and tourist behavior from 2015 to 2025? " The results indicate that all included studies were high-quality articles. The research characteristics of each paper were summarized, and overall publication volume trends were examined. Publication volume exhibits an overall upward trend, while journal publications demonstrate the advantages of interdisciplinary convergence and development.

### 3.3 Characteristics of space syntax indicators

Space Syntax is a theoretical and methodological framework for analyzing the relationship between spatial structures and human behavior [43]. Spatial configuration metrics derived from Space Syntax modeling, such as those based on axial maps and segment maps, primarily include integration, connectivity, depth, control, mean depth, and relative asymmetry [44], these metrics focus on spatial accessibility and connectivity. Visual features mainly include visual connectivity, visual integration, visual selectivity, visible area, and visible boundary length [45], these are based on the VGA (Visual Area Gauge) in spatial syntactic modeling methods, and these metrics focus on visual accessibility and visual perception. Therefore, both theoretically possess topological properties, but there are certain differences in their spatial syntactic modeling methods. The 16 articles included in this study cover topics such as depth value, integration, control value, selectivity, connectivity, visible area, visual integration, and composite indicators. To further understand the core indicators of Space Syntax, this study integrates spatial attributes and their metric characteristics to provide a detailed elaboration on spatial structure and accessibility, local control and path flow, local connectivity, spatial intelligibility, isovist area and visual integration, as well as extended metrics.

**3.3.1 Core space syntax metrics.** Within the framework of Space Syntax theory, depth is used to analyze the topological, metric, and angular relationships between spatial units, and it is generally categorized into three types: Step Depth, Metric Depth, and Angular Depth. Step Depth is calculated based on the number of steps along a path and reflects the hierarchical relationships within the overall structure of spatial units; Metric Depth is computed using the geometric length of space and emphasizes the actual physical distance and walking cost in space; Angular Depth is calculated mainly based on changes in turning angles along the path and emphasizes people's perception of the number of turns and the magnitude of turning angles [46]. This study uses Step Depth analysis to investigate historical gardens in which surveying accuracy is limited but spatial progression is emphasized.

In Space Syntax, the core indicators include depth, integration, connectivity, choice, control, and intelligibility (Table 6). Hillier and Iida [46] pointed out that Depth refers to the minimum number of steps required to move from one space to another. The smaller the Depth is, the more favorable the spatial location of that space. For integration, Gomaa et al. [33] further distinguish between "global integration'' and "local integration''; local integration measures the accessibility of a spatial unit within a specified range, that is, the node's accessibility within the local network, whereas global integration measures the accessibility of a spatial unit within the entire network. Connectivity is used to measure the number of adjacent spaces. The higher the Connectivity, the closer the relationships with adjacent spaces, indicating better spatial flow and more convenient traffic [30]. Control is calculated based on Connectivity and is used to evaluate the degree to which a spatial unit dominates its adjacent spaces. The higher the Control, the more pedestrian flows pass through the path entrances, increasing the likelihood of local pedestrian flows [39]. Choice is mainly used to measure the core position of spatial nodes; the smaller the Choice, the stronger the spatial centrality [46].

Mohammadi and Ujang [37] point out that intelligibility measures the degree of association between local space and the overall space, which reflects an individual's level of understanding of the overall spatial structure when within a bounded space. It reflects the extent to which local space facilitates individuals' understanding of the whole, revealing how movement and perception within space influence cognition of the spatial environment. The $R$ value represents the association between connectivity and integration(Table 7).

**3.3.2 Isovist area, visual integration and extended metrics.** Isovist Area refers to the spatial extent that can be included within an individual's field of view when standing at a given position. The more the surrounding space is covered, the larger the Isovist Area and the stronger the openness and permeability of that space [49].Yu et al. [32] further found that higher isovist values correspond to greater spatial transparency, and spaces with higher visual integration tend to exhibit higher visual accessibility and spatial guidance. Visual integration is a space syntax metric derived from Visibility Graph Analysis (VGA) that measures a spatial unit's accessibility and centrality within the visual network. Higher visual integration indicates greater spatial guidance and attractiveness. In garden environments, spaces with high visual integration are generally located in areas with strong intersect visibility, meaning that the view is less obstructed and the field of vision is open, which is more conducive to promoting social interaction and the flow of people.

Extended metrics refer to composite measures derived from space syntax core metrics (integration, connectivity, depth) in combination with other data or analytical tools. Among the 16 included studies, common research methods included combining GPS with Baidu heatmaps. Specifically, combining Baidu heatmaps can be used to reveal the impact of spatial

**Table 6. Characteristics of core Space Syntax metrics.**

| Space syntax measures | Formula | Feature description | Source |
|---|---|---|---|
| Depth | $MD_i = \frac{\sum_{j=1}^{n} D_{ij}}{(n-1)}$ | $MD_i$ represents the mean Depth of unit $i$, $D_{ij}$ represents the topological distance from unit $i$ to unit $j$, n is the total number of units in the space, and $n-1$ is the number of units excluding unit $i$. | Freire de Almeida et al.(2021) [47] |
| Integration | $I = \frac{(n-1)}{\sum_{j=1}^{n-1} d_{ij}/(n-1)}$ | $n$ represents the total number of nodes in the spatial unit, and $d_{ij}$ represents the number of steps in the shortest path from spatial unit $i$ to spatial unit $j$. | Lyu et al.(2025) [48] |
| Control | $CV(i) = \sum_{j \in N(i)} \frac{1}{deg(j)}$ | $CV(i)$ represents the Control value of spatial unit $i$, $N(i)$ is the set of all units adjacent to $i$, and $deg(j)$ represents the connectivity of the adjacent unit $j$. | Lyu et al.(2025) [48] |
| Choice | $Choice(i) = \sum_{s \neq i \neq t} \frac{\sigma_{st}(i)}{\sigma_{st}}$ | $\sigma_{st}$ represents the total number of shortest paths from node $s$ to node $t$, $\sigma_{st}(i)$ represents the number of those shortest paths that pass through node $i$. | Freire de Almeida et al.(2021) [47] |
| Connectivity | $C_i = \sum_{j=1}^{n} a_{ij}$ | $n$ denotes the total number of nodes in the spatial network; if spatial units $i$ and $j$ are directly connected, $a_{ij} = 1$; and $a_{ij} = 0$ otherwise. | Freire de Almeida et al.(2021) [47] |
| Intelligibility | $Intelligibility = R^2 = [Corr(C_i, I_i)]^2$ | The higher the Intelligibility value is, the clearer the spatial structure; the lower the value is, the more likely it is to cause a sense of disorientation. | Lyu et al.(2025) [48] |

**Table 7. Range of intelligibility values.**

| Level of intelligibility | $R^2$ Range | Characteristic description | Impact |
|---|---|---|---|
| High intelligibility | $R^2 \geq 0.70$ (0.70–1.00) | Local spatial characteristics accurately reflect the spatial structure. | High spatial accessibility |
| Moderate intelligibility | $0.4 \leq R^2 < 0.70$ | Local spatial characteristics are correlated with the overall structure but cannot fully reflect the spatial structure. | Spatial accessibility requires external assistance. |
| Low intelligibility | $R^2 < 0.4$ (0–0.40) | Local space is uncorrelated with the overall structure. | Low spatial accessibility. |

structure on tourist clustering distribution. For example, Huang and Lee [28] explored the combination of Baidu heatmaps and spatial syntax, finding a significant correlation between integration degree and tourist clustering heat.

Zhang et al. [29] combined spatial syntax with GPS trajectory data to generate specific indicators such as visit rate, average dwell time, average walking speed, and revisit rate using GPS data, they then used Spearman correlation analysis to examine the relationship between GPS indicators and spatial syntax indicators. The results showed that walkable accessibility determines the likelihood of a visitor's first visit, while visual features have a greater influence on a visitor's willingness to revisit. By integrating space syntax metrics with behavioral or perceptual data, extended metrics can systematically reveal associations between spatial characteristics and outcomes such as visitor clustering, satisfaction, and revisit intention [30]. They overcome the limitations of single metrics and, through multi-dimensional data integration, enhance the explanatory and predictive power for the relationship between garden spatial structure and tourist behavior.

### 3.4 The influence of garden spatial configuration on tourist behavior

The spatial configuration of gardens significantly influences tourists' path preferences and movement patterns, and exerts clear effects on their staying preferences, dwelling choices, and perceptual experiences [50]. Although space syntax metrics have rigorous mathematical definitions and computational formulas, their values lack universal, fixed thresholds and are typically require interpretation after normalization. Therefore, (Table 8) summarizes reference numerical values for a series of core metrics, including Integration, Choice, and Connectivity, to enhance understanding.

Differences in garden spatial structure cause tourists to exhibit different behaviors, mainly affecting tourist aggregation, stay hotspots, social tendencies, and path choices. Among these, path choice has the most significant impact on tourist behavior, especially in areas with high Integration, Zhai et al. [22] found that, in urban parks, paths with higher integration are chosen by tourists with significantly higher frequency. Similarly, Lee [34] supports this view in his study, noting that the level of integration is positively correlated with the frequency of tourist path choices. Tourists tend to favor areas with high spatial accessibility, indicating that highly integrated regions are more likely to attract tourist clusters. With respect to initial visit frequency, Zhang et al. [29] further found, based on GPS trajectory data, that tourists stay in hotspots closely coinciding with the distribution of visual integration; Pedestrian accessibility influences the frequency with which tourists first enter a given area, thereby shaping their length of stay and spatial preferences. Similarly, regarding the main entrance to the garden, Gomaa et al. [33] further found that the step depth of the primary entrance can effectively reflects differences in tourists' accessibility within the garden. For example, gardens with multiple main entrances can increase the frequency of tourist visits, thereby enhancing overall accessibility. Moreover, open spatial nodes within the garden tend to exhibit stronger visual connectivity. Interestingly, Yu et al. [32] hold a similar view and point out that nodes with higher visual integration concentrate the majority of tourist stay behaviors, indirectly indicating that within these spaces tourists are more inclined to engage in clustered social interaction, photography, and experiential activities. Additionally, Wu et al. [30] found that spatial nodes with higher connectivity tend to exhibit higher densities of tourist aggregation and thus function as "focal spaces", where tourists are more inclined to stay and appreciate plants, water features, rockeries, and sculptures; moreover, these spaces are characterized by more frequent social interaction and rest activities. Meanwhile, Mohamed et al. [39] emphasized that garden spaces with lower depth are generally situated at the margins of the layout and have

**Table 8. The influence of space syntax metrics on tourist behavior.**

| Space Syntax Metrics | Range | Evaluation criteria (relative values) | Behavioral effects | Author (Year) |
|---|---|---|---|---|
| Integration | Commonly 0–1 or 0–2 after normalization. | Higher than the system mean = high integration; lower = low integration. | High values: central, highly accessible, potentially attractive movement corridors; low values: peripheral, poorly accessible. | Zhai et al. (2018) [22]; Lee (2021) [34]; Huang & Lee (2023) [28]; Zhang et al. (2020) [29]; Yu et al. (2021) [38]; Wu et al. (2025) [30]; Traunmüller and Zarghami (2023) [41] |
| Choice | Normalized: 0–1; non-normalized varies with network size. | Top 10–20% by quantile considered high-choice main corridors. | High values: must-pass/backbone corridors; low values: branch routes. | Mohamed et al. (2023) [39]; Zhai et al. (2018) [22]; Gomaa et al. (2024) [33]; Wu et al. (2025) [30] |
| Connectivity | Number of directly adjacent nodes, typically 1–10+ | Higher than the system mean = strong connectivity. | High values: intersections/hubs; low values: dead ends | Wu et al. (2025) [30]; Gomaa et al. (2024) [33]; Lee (2021) [34] |
| Control | Influenced by the sum of the reciprocals of adjacent nodes' degrees. | Higher than the system mean = strong control | High values: intersections/squares; low values: edges/dead ends | Yu et al. (2016) [32]; Wu et al. (2025) [32]; Mohamed et al. (2023) [30] |
| Step depth (from main entrance)/ MD | Average step distance to the main entrance or other nodes | Less than the system mean depth = central; greater = peripheral | High depth: poor accessibility, avoidance; low depth: high permeability | Gomaa et al. (2024) [33]; Huang & Lee (2023) [28]; Traunmüller and Zarghami (2023) [41] |
| Isovist Area | Related to field-of-view openness; measured in area units | Higher than the system mean = transparent/open | High values: exploration and clustering, spectatorship; low values: constrained/hidden. | Chen et al.(2025) [42]; Chen & Yang (2023) [31] |
| Visual Integration | 0–1 normalization | Higher than the system mean = strong visual guidance | Visual hotspots, dwell points, wayfinding | Yu et al. (2016) [32]; Yu et al. (2021) [38]; Zhang et al. (2019) [40]; Wu et al. (2025) [30] |
| Extended Metrics | No universal range; used in combination with external data. | Compared with GPS/heatmaps/questionnaires | Enhance the explanation of satisfaction, revisit intention, restoration, and social interaction. | Zhang et al. (2020) [29]; Huang & Lee (2023) [28]; Saadativaghar & Zarghami (2023) [35]; Mohammadi & Ujang (2021) [37] |

NOTE: Both measures can be derived from visibility graph analysis (VGA), but they differ conceptually and computationally. Connectivity measures intervisible points, while Isovist Area measures continuous visible space.

reduced accessibility, which suppresses tourists' exploratory behavior, diminishes their willingness to enter these paths, and consequently leads to a gradual decline in visit frequency.

At the level of garden spatial perception, Yu et al. [32] further found that garden spaces with an configurations and relatively short viewing distances are more likely to stimulate tourists' willingness to explore freely. For example, lingering on waterfront platforms and in pavilions or corridors often leads to higher levels of social behavior and cultural interaction, helping tourists construct a more complete spatial interaction pattern. On the other hand, some empirical results indicate that the use of composite indicators can enhance the explanatory power of spatial analysis. Chen and Yang [31] indicate in their study that winding, undulating narrative paths can shape tourists', "perceptual rhythm" and "emotional engagement," thereby stimulating their exploratory desire and curiosity and enabling them, as they move along these narrative routes, to experience a richly layered spatial experience. Huang and Lee [28] further combined space syntax with Baidu heat maps and found that the degree of integration significantly influences tourists' spatial satisfaction and their intention

to revisit. Tourists' subjective spatial preferences were also shaped by integration levels. They were more likely to linger and wait for companions in areas with higher integration.

### 3.5 Tourist perception as a mediating effect

Among the 16 reviewed studies, tourist behavior is not entirely determined by the spatial structure of the garden itself. Tourists' subjective perceptions are an indirect factor influencing behavioral outcomes such as dwell time, movement speed, and social interaction, and are analyzed via perceptual variables including aesthetic preferences, cultural cognition, safety, and restoration [51] (Table 9). Zhang et al. [29] pointed out that the spectatorship of garden nodes engenders a "view-at-every-step" aesthetic perception; the revisitation rate increases significantly and average walking speed decreases, and when directional perception is clearer, route choice becomes more stable. Yu et al. [38] further support this view: visual perceptions such as coherence, openness, and legibility can significantly predict dwell density, evoke emotional experiences including immersion, mystery, safety, and imageability, and promote prolonged dwelling and contextual immersion. Chen and Yang [31] explore the mediating role of cultural symbols, arguing that narrative elements such as plaque inscriptions, couplets, and poetic paintings serve as carriers that more easily evoke tourists' understanding and experience of the "garden within a garden," storyline, thereby strengthening visual orientation and deepening overall spatial memory. Tourists' restorative perception are directly reflected in their behavior during their stay, suggesting that cultural symbols in garden spaces can enhance tourists' perceptual capacity. More notably, Mohammadi and Ujang [37] found that cultural landmarks and activity nodes help enhance tourists' sense of social safety, thereby promoting social interaction and stay behavior. Given the reciprocal relationship between cultural symbols and tourist perceptions, understanding the interaction between cultural connotations and spatial ambience remains an important direction for future research. For example, it is worth exploring how variations in spatial ambience influence tourists' physical and mental health and attention restoration. On the other hand, tourist perception can facilitate psychological restoration, and individuals exhibit varying levels of restorative response to different natural environments. Saadativaghar and Zarghami [35] show that changes in emotional dimensions are significantly associated with spatial configuration and recommend that garden layouts should alleviate crowded spaces and optimize disorganized spatial arrangements to enhance tourists' perceived mental health. In summary, tourists' aesthetic, aesthetic, cultural, safety, and restorative perceptions mediate the relationship between garden spatial configuration and behavioral outcomes. These perceptual factors influence processes such as wayfinding, path choice, length of stay, and social interaction, thereby indirectly shaping tourist behavior patterns and laying the groundwork for constructing a subsequent "Structure–Perception–Behavior" framework.

## 4. Discussion

### 4.1 The influence of garden spatial configuration on tourist behavior

Spatial configuration has a significant predictive effect on tourist behavior, with highly integrated paths often serving as primary circulation routes where tourists are more likely to congregate [52](Table 10). Zhai et al. [22] demonstrated that

**Table 9. The impact of tourist perception on behavioral outcomes.**

| Perceptual mediation | Spatial configuration | Perceptual variables | Behavioral outcomes | References |
|---|---|---|---|---|
| A. Aesthetic preferences | Visual focus/ legibility/ accessibility/ centrality | Enclosure/ mystery/ explorability/ spectatorship/ view framing/ borrowed scenery | Dwell density ↑revisitation ↑ speed ↓ | Zhang et al. (2019) [40] Yu et al. (2021) [38] Chen et al.(2025) [42] Zhai et al. (2018) [22] |
| B. Cultural cognition | Salience of cultural landmarks | Cultural symbolism/ narrative understanding/ imageability consistency | Dwelling at cultural nodes↑ Visual guidance enhanced spatial memory | Chen & Yang (2023) [31] Yu et al.(2016) [32] |
| C. Restoration/ safety/ social interaction | Depth/ concealment/ visibility | Restorativeness/ comfort/ friendliness | Path preference↑ Dwelling and social interaction↑ Restoration score↑ | Saadativaghar and Zarghami. (2023) [35] Mohammadi & Ujang (2021) [37] |

**Table 10. The relationship between garden spatial configuration and tourist behavior.**

| Author (Year) | Sample | Behavior Variable | Limitations |
|---|---|---|---|
| Zhai et al. (2018) [22] | Main pathways | Route choice and accessibility | Lacks micro-level dwell behavior and individual differences |
| Wu et al. (2025) [30] | Node network | Interpersonal interaction | Lack of linkage between behavioral and perceptual variables |
| Mohammadi & Ujang (2021) [37] | Path network | Frequency of social interactions | Missing contextual variables (seating, commercial services, security) |
| Zhang et al. (2020) [29] | VGA grid cell | Dwell hotspots | Single behavior type; social interaction not addressed |
| Chen & Yang (2023) [31] | Narrative unit | Perception of cultural imagery | No behavioral data collected; high subjectivity |
| Lee (2021) [34] | Garden entrances and pathways | Visitation frequency | Lack of stratified analysis of tourist types and subjective perceptions |
| Mohamed et al. (2023) [39] | Observed pedestrian flow | Route choice | Did not consider cross-cultural differences and cultural nodes |

spatial accessibility is highly correlated with the integration and that integration predicts path choice. However, analysis was limited to the garden's main routes, lacking examination of stay behavior at the micro level and failing to consider differences in tourists' individual characteristics. As a result, the findings did not reflect how spatial configuration influences tourists' emotional experiences and cultural understanding. Wu et al. [30] demonstrated that higher depth and intelligibility are associated with greater accessibility of spaces for tourists. However, their analysis was confined to VGA and segment angular analysis, without collecting empirical data on tourists' actual behavior. Future research should therefore integrate observed tourist behavior and conduct more in-depth investigations of actual movement data. In addition, Mohammadi and Ujang [37] examined the relationship between spatial path accessibility and the frequency of social interaction, confirming that nodes with high accessibility and connectivity are more likely to serve as social focal points. However, the study did not fully explore external factors linking tourist behavior and spatial configuration, such as seating provision, commercial facilities, and security installations, which also play an essential role in interaction frequency.

Existing studies still have certain limitations in methodology and research subjects. Zhang et al. [29] collected actual movement data from 353 tourists using recorders and confirmed that spatial visual characteristics are more likely to attract higher visit frequencies, proposing that pedestrian accessibility determines tourists' initial visit frequency. However, the study area was limited to the Lion Grove Garden, and the data collection period did not cover either winter or summer, resulting in a lack of behavioral comparisons across seasons to verify the reliability of the findings, and leading to a relatively homogeneous behavioral pattern. Although Chen and Yang [31] although combined Space Syntax with spatial narrative theory and emphasized the guiding role of spatial focal points and cultural imagery in shaping tourists' emotions, they did not collect empirical data on tourist' behavior, thereby limiting the study's objectivity. Future research should incorporate GPS data to achieve a more comprehensive understanding of how spatial configuration influences tourist behavior. Cross-cultural applicability remains a weak point in current research. For example, Mohamed et al. [39] focused on the relationship between garden path connectivity and visitor behavior but failed to consider the regulatory role of cultural background in shaping tourist actions.

Overall, integration, connectivity, and visual integration exhibit significant explanatory power for tourists' routing and social behavior. In particular, nodes with high integration and high connectivity often become spatial focal points for visitor aggregation and interaction. Nevertheless, several limitations remain: (1) most study objects are concentrated in restricted garden types, with a lack of systematic comparison at the micro level regarding subjective perception and cultural behavioral differences; (2) insufficient control of external facilities and management conditions, with contextual variables inadequately incorporated; and (3) cultural and cross-cultural differences were not analyzed as variables.

## 4.2 Tourist perception as a mediating effect

This review further reveals that tourist perception serves as a mediating factor between spatial structure and behavioral patterns, thereby constructing a "structure–perception–behavior" pathway. Nevertheless, most existing studies have conducted only quantitative analyses of spatial characteristics and have failed to incorporate dynamic influencing factors, such as pedestrian flow, climate, aesthetic preferences, and temporal variation. Chen and Yang [31] found that spatial narratives enhance tourists' emotional immersion and cultural associations. However, their study lacks quantitative evidence on how these cultural associations translate into behavioral responses. Therefore, although existing studies have begun to reveal the mediating role of tourist perception, the analysis of perceptual dimensions remains incomplete, data collection is unsystematic, and research methods are inconsistent. As a result, the mediating effect has not yet been systematically modeled. Future research should aim to deepen the theoretical construction and empirical testing of this mediating mechanism by employing standardized measurement scales and developing structural equation models. (Fig 5) visually presents the pathways between space syntax metrics and tourist perception and behavioral outcomes in existing studies, laying the groundwork for the "Structure–Perception–Behavior" framework proposed in the next section.

## 4.3 Structure–Perception–Behavior (SPB) framework

Based on a systematic review of the relevant literature, and drawing on Mehrabian and Russell's Stimulus–Organism–Response (SOR) model and Rapoport's Culture–Environment–Behavior (CEB) explanatory framework, this study clarifies the "X→M→Y" mediating model mechanism and constructs a "structure–perception–behavior" (SPB) framework [53,54] (Fig 6). The framework emphasizes that spatial structural features can directly influence tourists' route choice and behavioral performance through space syntax metrics such as integration, connectivity, depth, visual integration, and isovist area. Meanwhile, tourist perception (aesthetic preferences, cultural cognition, sense of safety) plays a key mediating role between physical space and behavioral performance. Spaces with high integration typically exhibit greater visual centrality and spatial accessibility. Their clear orientation and easily recognizable routes enhance the spatial configuration's

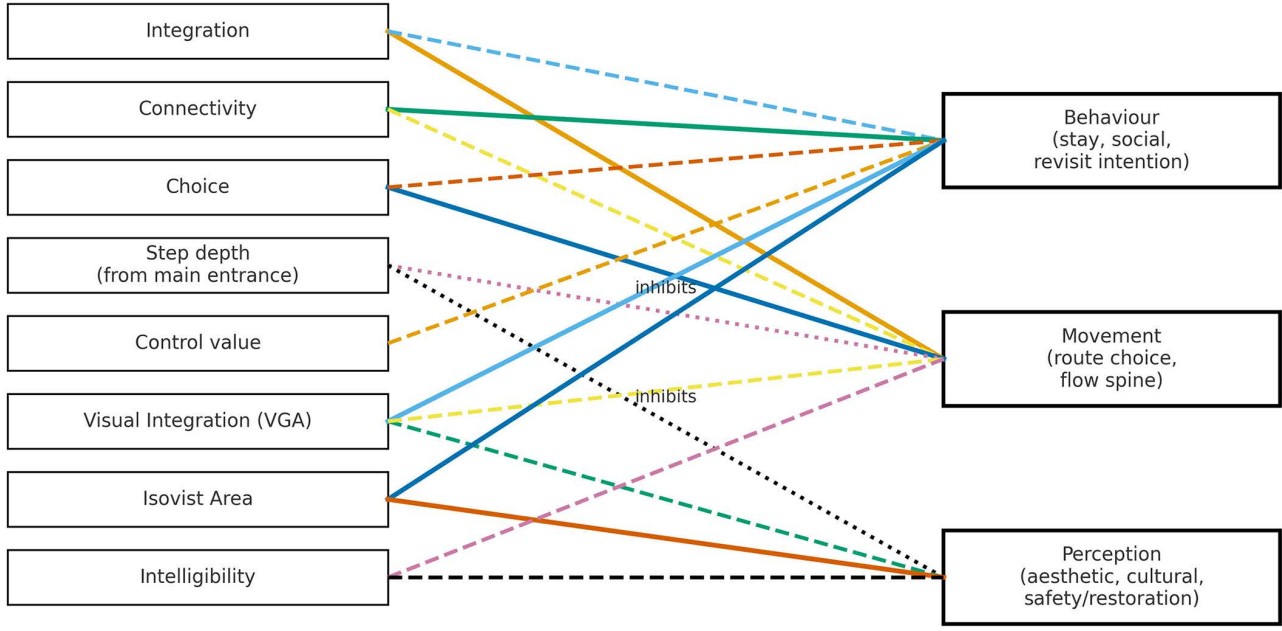

**Fig 5. Space syntax and tourist outcomes.**

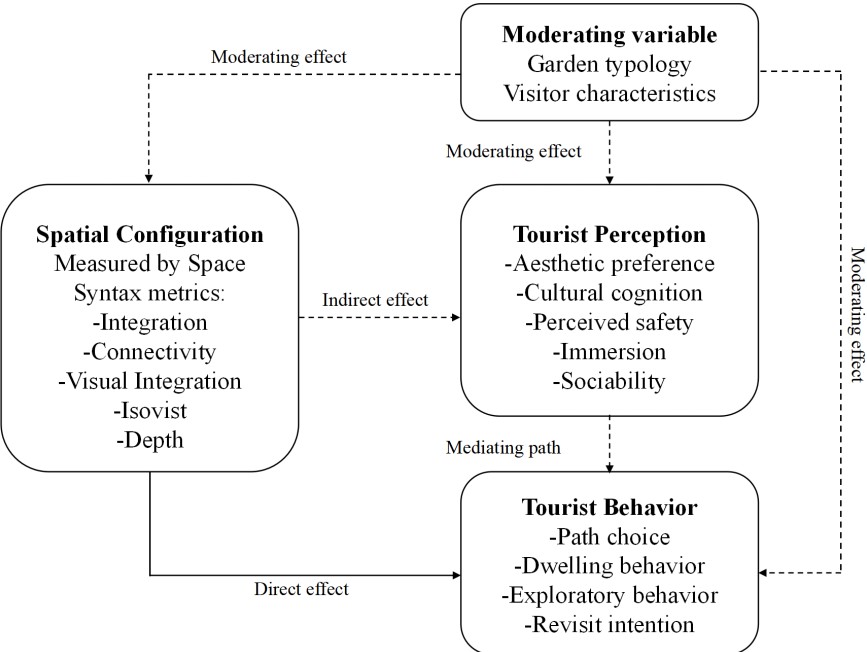

**Fig 6. Structure–Perception–Behavior (SPB) framework.**

identifiability and legibility. Accordingly, the more recognizable and legible a space is, the higher its integration tends to be. In addition, high Choice increases the likelihood that a path functions as a primary route, and the formation of primary routes, in turn, reinforces high Choice. Through this mutual influence, high-choice main routes often become centers of human activity and aggregation and are more prone to high pedestrian flow and congestion. Consequently, Choice is positively associated with behavioral outcomes: the higher the Choice value, the greater the probability that a path will be traversed, thereby shaping tourists' route preference decisions.

Additionally, this study incorporates garden type and tourist characteristics as moderating variables, enabling the conceptual framework to compare differences across cultural backgrounds and sociodemographic attributes. The significance of this framework lies in its integration previously fragmented empirical findings into a testable theoretical model, providing a foundation for further quantitative validation and thereby advancing systematic research on the relationship between garden spatial configuration and tourist behavior.

### 4.4 Cross-scale methodological implications for garden spatial research

This review centers on garden spatial configuration, emphasizing gardens space as a typical form of "small-scale urban space." By comparison, how other urban spaces at different scales influence behavioral outcomes (Table 11) provide essential references and insights for research on gardens.

At the architectural and interior scales, VGA is widely used to reveal local visibility and patterns of occupancy. In office settings, employee interaction frequency in high-integration areas is 2–3 times higher than in low-integration areas [55]. In school buildings, the integration of primary corridors is 30%–50% higher than that of secondary corridors, exerting a significant influence on students' route choice and space use [56]. In hospital settings, the visibility of wards and corridors is directly related to staff rounding efficiency and patient accessibility [57]. In museum spaces, the visual integration of galleries is highly correlated with visitors' tour routes and dwell hotspots [58]. At the neighborhood and city scales, studies

**Table 11. Comparison of urban spaces and behavioral outcomes across different scales of study.**

| Author (Year) | Study objects | space syntax modeling methods | Main findings | Data support | Implications for garden research |
|---|---|---|---|---|---|
| Peponis et al. (1990) [55] | Office buildings | VGA | High-integration areas exhibit significantly higher interaction frequencies than low-integration areas. | In high-integration areas, interaction frequency is 2–3 times higher than in low-integration areas. | In high-integration areas, interaction frequency is 2–3 times higher than in low-integration areas. |
| Hillier (1996) [56] | School | VGA | Student path choice is closely related to the integration of corridors/classrooms. | Primary corridors exhibit 30%–50% higher integration than secondary corridors. | Garden visitors tend to choose high-integration paths as their routes. |
| Haq & Zimring (2003) [57] | Hospital | VGA | The visibility between wards and corridors influences rounding efficiency and patient accessibility. | High-visibility areas can shorten rounding routes by 25–30% and improve accessibility. | High-visibility spaces in gardens help enhance accessibility and efficiency. |
| Choi (1999) [58] | Museum | VGA | Exhibition-hall visual integration is highly correlated with visitor routes and dwell hotspots. | In high-integration galleries, visitor dwell time is 1.5–2 times longer, and visitation is more concentrated. | In gardens, high-integration areas are often hotspots for visitor congregation and dwelling. |
| Hillier & lida (2005) [59] | London streets | Segment (Axial/Angular) | Global integration is significantly associated with pedestrian flow patterns. | Correlation coefficient between pedestrian flow and integration $R^2 \approx 0.6$–$0.7$ | Research on garden path structures can be linked to comparative analyses of urban travel patterns. |
| Jiang & Claramunt (2002) [60] | French streets | Segment | Street connectivity is highly correlated with traffic volume. | On high-connectivity roads, traffic volume is 40–60% higher than on low-connectivity roads. | In gardens, highly connected paths often serve as primary corridors for visitor movement. |

commonly employ segment analysis to explain traffic flows and block connectivity. The correlation between global street integration and pedestrian movement patterns can reach $R^2 \approx 0.6$–$0.7$ [59]. Further studies indicate that traffic volumes on highly connected streets are 40%–60% higher than on low-connectivity streets [60].

These studies indicate that the space syntax approach can still accurately analyze individual behavioral responses in urban spaces of varying scales (office buildings, schools, hospitals, and streets). This not only reinforces the appropriateness of applying space syntax methods to "small-scale urban space (garden spaces)," but also provides a practical foundation for future cross-scale comparative research.

## 5. Limitations

In terms of literature retrieval, although this study searched four major platforms for published studies and sought to cover research on space syntax and gardens comprehensively, some omissions remain unavoidable. Due to the indexing mechanisms and language limitations of different databases, specific grey literature with potential research value is challenging to obtain, and exceptionally high-quality studies published in other language systems that are hosted on regional databases cannot be included. Although this study applied the PICOS criteria and the CCAT tool to conduct a rigorous qualitative appraisal of the included studies to compensate for the limitation of a relatively small sample size, the robustness of the findings is still affected to some extent. Moreover, given differences in spatial scales and modeling methods, the present study confines its discussion to the relationship between small-scale garden spaces and tourist behavior, thereby limiting cross-scale comparisons. While typical application scenarios at other scales, such as building interiors, urban streets, and campus or park environments, are briefly mentioned, the overall analytical scope remains bounded by the garden scale, and the conclusions cannot yet be readily generalized to other spatial scales. In addition, this study primarily focuses on Chinese classical gardens and certain urban gardens in Asian cities, thereby limiting the generalizability of the results. Therefore, future research could expand the range of multilingual retrieval platforms and include a broader set of studies to enlarge the sample coverage and enhance the robustness of the evidence base.

## 6. Conclusions

To elucidate the multiple influences shaping tourist behavior, this study systematically examines the relationship between garden spatial configuration and tourist behavior through a literature review. The findings show that existing research employing space syntax to explore this relationship has predominantly focused on East Asian traditional gardens, and that the number of such publications has increased markedly in recent years. To further clarify the core metrics of space syntax, this study integrates spatial properties with metric characteristics. It provides a detailed discussion of spatial configuration and accessibility, local control and path flow, regional connectivity, spatial intelligibility, isovist area, and extended measures. It argues that key space syntax measures such as Integration, Connectivity, Depth, Isovist area, and Visual integration have significant explanatory power for tourist behavior, and that nodes with high Integration and Connectivity tend to become spatial focal points for tourist aggregation and interaction. Meanwhile, by analyzing perceptual variables such as aesthetic preferences, cultural cognition, and a sense of safety, this study argues that tourist perception mediates between spatial configuration and behavioral patterns and accordingly develops a Structure–Perception–Behavior (SPB) framework. In doing so, the study not only integrates key findings from existing research at the local level but also, at an overall level, provides a theoretical framework and methodological support for understanding the multiple ways in which garden spaces influence tourist behavior. Finally, this study extends the discussion from garden spaces to building and interior spaces, urban blocks, and the city as a whole, and proposes cross-scale methodological implications. Overall, it not only demonstrates the applicability of space syntax in garden planning and tourist behavior research, but also points to directions for future work to construct empirical models that verify the proposed mediating mechanisms, thereby enhancing both the breadth and theoretical depth of research in this field.

## Supporting information

**S1 Table. Detailed search strategy.**
(PDF)

**S2 Table. Crowe Critical Appraisal Tool (CCAT) form.**
(PDF)

**S3 Table. PRISMA 2020 checklist.**
(PDF)

**S4 Table. Numbered table of all studies.**
(PDF)

## Author contributions

**Conceptualization:** Yan Han, Jasmine Leby Lau.

**Data curation:** Bin Li, Yan Han, Jasmine Leby Lau.

**Formal analysis:** Bin Li, Yan Han.

**Investigation:** Bin Li, Yan Han.

**Methodology:** Bin Li, Yan Han.

**Project administration:** Mohammad Mujaheed Hassan, Jasmine Leby Lau.

**Resources:** Bin Li, Mohammad Mujaheed Hassan, Yan Han.

**Software:** Bin Li, Yan Han.

**Supervision:** Mohammad Mujaheed Hassan.

**Validation:** Bin Li, Mohammad Mujaheed Hassan.

**Visualization:** Bin Li.

**Writing – original draft:** Bin Li, Yan Han.

**Writing – review & editing:** Bin Li, Yan Han.

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
