## [Decision Letter · Decision Letter 0]

25 Aug 2025

Dear Dr. %:LAST_NAME%,

Thank you for submitting your manuscript to PLOS ONE. After careful consideration, we feel that it has merit but does not fully meet PLOS ONE’s publication criteria as it currently stands. Therefore, we invite you to submit a revised version of the manuscript that addresses the points raised during the review process.

We look forward to receiving your revised manuscript.

Kind regards,

Yile Chen, Ph.D. in Architecture

Academic Editor

PLOS ONE

Journal Requirements:

2. Thank you for stating the following in your Competing Interests section: [No]

Reviewers' comments:

Reviewer's Responses to Questions

**Comments to the Author**

1. Is the manuscript technically sound, and do the data support the conclusions?

Reviewer #1: Yes

Reviewer #2: Yes

Reviewer #3: Partly

Reviewer #4: Yes

2. Has the statistical analysis been performed appropriately and rigorously?

Reviewer #1: Yes

Reviewer #2: Yes

Reviewer #3: N/A

Reviewer #4: Yes

3. Have the authors made all data underlying the findings in their manuscript fully available?

Reviewer #1: Yes

Reviewer #2: Yes

Reviewer #3: Yes

Reviewer #4: Yes

4. Is the manuscript presented in an intelligible fashion and written in standard English?

Reviewer #1: Yes

Reviewer #2: Yes

Reviewer #3: Yes

Reviewer #4: Yes

Reviewer #1: 1.While the conceptual pathway is sound, the review relies heavily on synthesizing prior findings without clearly proposing or testing a new theory. The model would benefit from a visual diagram or conceptual map.

2.Despite following PRISMA guidelines and employing the CCAT tool, the scope of analysis is narrow. Only 10 studies are included over a 9-year span. The justification for this small sample should be more robust (based on my knowledge.)

3.The synthesis could benefit from clearer distinction between quantitative behavioral data (e.g., GPS paths, density) and subjective perceptual data. Future reviews could adopt mixed-method meta-analysis frameworks.

4.Some terms like 'isovist' or 'visual integration' should be briefly defined when first used.

5.Ensure figures referenced (e.g., PRISMA diagram, Fig. 3) are embedded correctly in final layout.

6.Writing is mostly clear but would benefit from further proofreading for redundancy and flow.

Reviewer #2: The manuscript examines a pertinent and timely subject with potential significance for both scholarly inquiry and practical implementation. The study design seems methodologically robust, and the data provided corroborate the overarching conclusions. The statistical analyses utilise suitable methods and seem to have been conducted with sufficient rigour, enhancing the reliability of the findings.

The manuscript is predominantly well-composed, with arguments articulated in a coherent and logical fashion; however, minor grammatical revisions and occasional rephrasing could enhance its flow and readability. Certain terms and definitions require more precise clarification to enhance accessibility for readers from varied backgrounds.

Although the methodology is valid, the authors should offer more explicit justification for specific decisions regarding data handling, analysis, or variable selection to enhance the robustness of their approach. It would be advantageous to succinctly emphasise the originality and prospective impact of this research in the introduction, clearly articulating how it enhances current understanding.

The manuscript possesses merit and, with minor enhancements in clarity and language refinement, can significantly contribute to the literature.

Reviewer #3: This is an interesting article that accumulates recent developments in the use of spatial configuration metrics to determine patterns in tourist behaviour in gardens. The authors appear to have done a relatively comprehensive collection, final selection and description of research studies relying on existing models and standard processes. The manuscript is readable and intelligible though slightly repetitive.

Where the article falters is in going beyond simple enumeration of the methods used in each study particularly as that relates to spatial configuration. While this is indeed an emerging domain with various published papers, the strategy for the review chosen here seems lacking. The authors have chosen to do a relatively shallow reading of the methodology of each article, focusing only on metrics, whereas research that is typically considered part of the space syntax domain has produced a multitude of combinations of spatial representations, algorithms, metrics and other methodological considerations such as radius (the spatial limit to the reach of the various algorithms). This approach would have worked fine if the sample was larger, but there are only 10 studies selected and all use rather different methods of getting to the result, not always equivalent.

Because the sample is small, I would have expected a much deeper explanation of the spatial configuration methodologies. For example, a cursory reading of the studies suggests that they use different representations of space, but these are not mentioned in the manuscript. They do not appear as part of the methodology review or identified per-study. Some studies seem to use Visibility Graph Analysis (VGA) (referenced as 13, 15, 30 and 31), some Segment maps (28, 30, 32, 33) some Isovists (27, 31) and some Convex maps (20, 29). This can be problematic in some cases because metrics such as "Integration" and "Connectivity" produce different values for different representations at the same location (especially VGA and Segment maps) making comparison to human behaviour and perception difficult. Given the small sample I would have also at least expected the authors to further provide examples of the use of these methods in other scales of the built environment. Gardens are at the "small urban area" scale, meaning that the authors could have examined papers from both smaller scales (buildings such as offices, schools and hospitals, usually employing VGA) and larger scales (large urban areas and cities typically employing segment analysis).

Instead, the authors demonstrate that the studies themselves have only had surface-level examination. This is evident from examples such as:

1. attributing analysis to Gomaa et al. (32) using "Local integration" whereas the paper in question only mentions the metric in passing in the introduction,

2. referring to "depth" as a metric, which is unusable on its own and requires a location descriptor (as in Mohamed et al. (2024)[29] actually referring to it as "Step depth from the main entrance")

3. having both "connectivity" (Wu et al. (2025) [30]) and "isovist area" (Yu et al. (2016) [31]) as different metrics where they are practically the same because the two studies use VGA,

4. mentioning "extended metrics" in Tables 6 and 7, attributing it to different papers but never explaining what those metrics are

5. completely ignoring other important metrics such as control found in Mohamed et al. (2024)[29] and choice found in Wu et al (2025) [30].

This shallow approach has also prevented the authors from drawing larger conclusions in their discussion such as how and, more importantly, why some metrics affect tourist behaviour. For example the authors do identify that Integration is an important metric but do not go further to try to identify why it might correlate with tourist path selection. Is it perhaps a strategy of people who don't know a space (tourists) to traverse that space on what looks to them as the most obvious path? Is it that this is the result of exploratory behaviour as in other cases (say galleries)? The authors instead avoid the larger critical reflections (which would be fitting for a such a review) and choose to recite the various results from the studies highlighting the limitations (which seem to also come from the studies).

Other notes:

The referencing is inconsistent to the point where it creates confusion:

1. One of the main studies "Zhang et al. (2020)" referenced as 15 has a broken DOI and either the wrong title or the wrong authors. The title matches authors Zhang T., Lian Z., and Xu Y and the DOI https://doi.org/10.1080/01426397.2020.1730775.

2. It is unclear what "extended metrics" are, they are attributed to either Lee, K. (2021) in Table 6 or Huang and Lee (2023) in Table 7

Reviewer #4: PLOS ONE, Manuscript Number: PONE-D-25-34363

Title: The Influence of Garden Spatial Configuration on Tourist Behavior: A Systematic

Review Based on Space Syntax

This article presents a systematic scoping review of space syntax research on the influence of garden spatial configurations on tourist behavior. Employing the PRISMA framework, the review identifies ten relevant studies from four major databases: Web of Science, Scopus, JSTOR, and ScienceDirect. The review first highlights research that utilized spatial-perceptual methods in garden analytics from 2016 to 2025. It then identifies key syntactic metrics that significantly contribute to understanding garden spatial configurations and explaining users’ navigational behavior. Furthermore, the review explores the perceptual experiences associated with garden spaces, emphasizing the need for a more integrated analytical framework - one that incorporates emotional-perceptual simulations and cultural symbolism, both of which indirectly influence tourist behavior. The study concludes by addressing existing research gaps and suggesting directions for future investigation.

The paper presents an interesting focus, and it has the potential to be publishable if significantly revised. However, it currently suffers from several issues that undermine its quality and usefulness to the reader. I encourage the authors to undertake a major revision to address these concerns, as the paper contains valuable insights that could be of interest to readers.

Reviewer comments

First,

The biggest issue with the paper is the unclarity of its questions. The authors fail to clearly highlight the specific research questions they themselves aim to sequentially address in this study – apart from the questions that were raised by the articles of the literature review. All of which confuses the reader.

As the authors emphasized that their main research interest was 'Examining the relationship between garden spaces and touristic behavior,' they posed two questions. While RQ1 is not clear in its academic intent, and could therefore be adjusted, RQ2 is more relevant but would benefit from being developed into several operational sub-questions (among which the modified/altered RQ1 could be incorporated). This lack of clarity in the formulation of the research questions directly affects the overall rigor of the study. In its current form, the paper does not adequately address the research operational questions, leaving the reader with a conclusion that offers little specificity regarding results relevant to the central focus. The final sections continue to reference existing literature rather than engaging with the study’s own findings. A stronger approach would have been to adopt a singular overarching focus (examining the relationship between garden spaces and touristic behavior), develop several operational sub-questions, rigorously investigate each of them, and then provide a conclusive answer supported by empirical evidence/finding. Although the authors have made considerable effort and generated the findings needed, the knowledge that they developed is not presented in a structured or clear way; important findings are thrown as familiar statements, and the headings and the sub-headings of the sections do not always hint to the link between the question inquired and the content presented. Based on a review of the current content of their study, the authors should formulate three to four operational questions and restructure the study to present clear, definitive answers.

In general, the authors need to revise the text, develop a multi-layered discussion, and clearly highlight their contributions at three levels: (1) data collection and PRISMA findings, (2) the application of space syntax metrics and how these metrics are used and analyzed to reveal specific configurational–behavioral experiences - in addition to the other supporting theories if any, and (3) the interpretations derived from the multi-layered study and the cumulative insights of the ten reviewed papers.

Second

The methodology outlining how the collected data is investigated, interpreted, and how each layer of the analysis and research questions is addressed needs to be presented more clearly, both in text and through a flow chart. While the authors included visual graphs and tables related to the literature review and its summary. However, in the sections /levels where their own input is most important and potentially influential, this impact is minimized. Summaries concluded section-paragraphs, interpretations, and visualizations should be used more effectively to emphasize their contribution and to clearly demonstrate the methodology applied in screening and analyzing (the content of) the selected papers, which is critical. A chart detailing this methodology is expected to be incorporated prior to the presentation of the results.

Furthermore, the Methods section unnecessarily expands on subsections 2.2 (Criteria) and 2.4 (Quality Assessment), which distracts the flow of the main core topic. It could be more effectively reduced and summarized in a simple paragraph and a table.

It should also be clarified that, while many studies employ similar large databases, the relatively small number of papers reviewed in this specific topic limits the extent to which statistical findings alone can be relied upon. This makes a deeper level of analysis and interpretation in this study particularly important.

Third

While Table 4. - which appears in section 3.1., provides a comprehensive summary and introduces several socio-perceptual and phenomenological tools/theories, it would be useful to include a paragraph(s) briefly describing the other tools (beyond space syntax) that are mentioned. In other words, authors need to present a brief explanation to the “theories” that appear in the column of “theory” Table 4., and to clarify how these relate, or not relate, to Space Syntax, and consequently the title of this study. (Examples TG, CL, NRT, IL, TB,….. and why they are included).

In addition, the meaning of 'Visitor involvement level' in Table 4 is unclear; its relevance to the investigation should be clarified; otherwise, it could be omitted.

Fourth,

Table 6. in Section 3.3 could be expanded by adding images of the layouts of the 10 cases studied with certain space syntax and visibility maps (with clear references). The visuals of the space syntax metrics as reflected in graphs usually help understanding the findings and help in understanding the context.

Fifth,

To improve the readability of Table 7 which appears in Section 3.4, it would be helpful to add a column entitled “Space Syntax Metrics” to clarify the metrics used in each of the studies listed. Alternatively, Tables 6 and 7 could be combined into a single table.

Sixth,

Section 3.5 is an important part of the paper as it highlights the distinction between the impact of garden spatial configuration and perceptual values such as aesthetic preferences, cultural cognition, and sense of safety. The authors should emphasize this section more clearly to underline their contribution. Including a table that summarizes these issues would also strengthen the presentation.

Seventh,

The presentation of the text in Section 4.1 gives the impression of abundance and unnecessary repetition. The authors should clarify the distinctions between the metrics, their analytical value, and the interpretive insights that emerge when different space syntax indicators are combined. Since the power of integrating tools is central to this section and represents the authors’ key contribution, it should not be downplayed by the inclusion of repeated content. Reference to the 10 studied papers could help (Optional).

Eighth,

The authors are advised to include a table at the end of Section 4.2, followed by an interpretive paragraph.

Ninth,

Section 4.4 would benefit from a concluding paragraph that summarizes the findings and highlights the authors’ interpretations, thereby underscoring their own voice and contribution.

Tenth,

The conclusions should directly address the research questions and provide clear answers at both the local and global levels.

Other issues:

One,

The following paper that you referred to in your study is an excellent example, particularly in how it demonstrates the alignment between paragraphs and figures (see, for instance, Figures 3 and 4): Lee, J.H.; Ostwald, M.J.; Zhou, L. (2023). Socio-Spatial Experience in Space Syntax Research: A PRISMA-Compliant Review. Buildings, 13(3), 644. https://doi.org/10.3390/buildings13030644

Two,

A note to look at:

At the 4th International Space Syntax Symposium in 2003, M. Abrioux presented an article entitled 'Body, eye and imagination: A meditation on the dynamics of space in French and English gardening.' The paper applied space syntax to garden analysis, emphasizing the perceptual and imaginative dimensions of spatial experience in French and English garden design. Although not a comprehensive study of gardens, it represents a key contribution in that year by extending space syntax concepts into the context of landscape architecture and highlighting how garden design can shape visitors’ perception and experience of space. The research likely focused on the subtle, qualitative aspects of spatial design rather than just quantitative metrics, which are more common in other space syntax applications. It might help to add this paper to your references for its direct relevance to the chronological literature review. Following are the details of the reference.

Abrioux, M. (2003). Body, eye and imagination: A meditation on the dynamics of space in French and English gardening. Paper presented at the 4th International Space Syntax Symposium, London.

**Do you want your identity to be public for this peer review?** For information about this choice, including consent withdrawal, please see our Privacy Policy

Reviewer #1: **Yes: ** Adib Amany

Reviewer #2: **Yes: ** SAYON PRAMANIK

Reviewer #3: No

Reviewer #4: **Yes: ** Shatha Malhis

---

## [Author Response · Author response to Decision Letter 1]

22 Sep 2025

Response to Reviewers' Comments

Title: The Influence of Garden Spatial Configuration on Tourist Behavior: A Systematic Review Based on Space Syntax

Manuscript ID: PONE-D-25-30762

We sincerely thank the academic editor and reviewers for their valuable feedback. We have addressed all comments and revised the manuscript accordingly. Our detailed responses are provided below.

Reviewer #1:

1.Comment:While the conceptual pathway is sound, the review relies heavily on synthesizing prior findings without clearly proposing or testing a new theory. The model would benefit from a visual diagram or conceptual map.

Response: Thank you for this helpful suggestion. We have added a visual framework to clarify the conceptual pathway and make the synthesis more transparent. The figure is placed at the end of the Introduction and labeled “Fig 1. Flow diagram for literature review.” It outlines the key constructs and their relationships and summarizes the review process. We believe this diagram strengthens the logic of the model and improves readability.

In addition, we have introduced a mechanism-oriented synthesis in Section 4.3 “Structure–Perception–Behavior (SPB) framework,” illustrated in “Fig 6. Structure–Perception–Behavior (SPB) framework.” This framework articulates how spatial structure (e.g., integration, connectivity, visual integration, isovist area) influences tourist perception (openness/readability, cultural cognition, safety/restoration), which in turn shapes behavioral outcomes such as path choice, dwell, and revisit intention.

2.Comment:Despite following PRISMA guidelines and employing the CCAT tool, the scope of analysis is narrow. Only 10 studies are included over a 9-year span. The justification for this small sample should be more robust (based on my knowledge.)

Response: Thank you for pointing this out. In response, we have re-screened the studies under Section 2.3. Study selection, and the final number of included studies has been updated to 18, in accordance with the predefined inclusion criteria. We have also revised the corresponding flow chart, now presented as “Fig 2. PRISMA flow diagram.”

Furthermore, in Section 2.2. Criteria and Quality assessment, we have added the following explanation:

“Although the search covered major databases, the number of studies ultimately included was relatively small; therefore, this study does not rely solely on statistical frequencies but adopts an interpretive synthesis, emphasizing the correspondence between ‘metrics–representations–behavior’ and the elucidation of the ‘structure–perception–behavior’ mechanism.”

We believe these revisions provide a more robust justification for the sample size and clarify the methodological rationale.

3.Comment:The synthesis could benefit from clearer distinction between quantitative behavioral data (e.g., GPS paths, density) and subjective perceptual data. Future reviews could adopt mixed-method meta-analysis frameworks.

Response: Thank you for this constructive comment. In the revised manuscript, we have provided a clearer distinction between quantitative behavioral data and subjective perceptual data. Specifically, in Section 3.1 “Research characteristics”, we added Table 4. Characteristics of the studies, which systematically summarizes the included studies according to different spatial representations (VGA, Segment, Isovist, Convex) and links them to different types of behavioral outcomes.

For subjective perceptual data, we added a dedicated subsection “3.5. Tourist perception as a mediating effect”, which systematically synthesizes perceptual variables such as aesthetic preference, cultural cognition, and safety/restoration. These perceptual variables are explicitly linked to spatial-structural drivers and behavioral outcomes. To complement this, we introduced Table 8. The impact of tourist perception on behavioral outcomes, which presents the relationships in a structured manner.

We believe these additions enhance the clarity of the synthesis and address the reviewer’s request.

4.Comment:Some terms like 'isovist' or 'visual integration' should be briefly defined when first used.

Response: Thank you for pointing this out. In the revised manuscript, we have added brief definitions of key terms when they first appear in Section 3.3 “Characteristics of space syntax indicators.” Specifically:

3.3.1. Depth and Integration

Specific content added: “Mohamed et al.[30] emphasize that depth refers to the total number of steps required from a spatial unit to other spaces, which can reveal the marginal characteristics of the spatial structure; depth is effective for assessing less-frequented areas…… Conversely, lower integration values indicate that the space lies in peripheral and relatively enclosed areas, with a higher degree of enclosure and concealment[45].”

3.3.2. Control value and Choice

Specific content added: “Mohamed et al.[30] indicate that the control value measures the relative control of a spatial node over its adjacent nodes within a spatial network, reflecting the functional mechanism of the focal spatial unit in the local spatial structure and serving to assess its effects on surrounding spatial units…...Higher choice values indicate that the spatial unit plays a central role in route selection within the overall network[45].”

3.3.3. Connectivity and Intelligibility

Specific content added: “Wu et al.[31] state that connectivity refers to the number of links from a given spatial unit to its adjacent units, reflecting local connectedness…….Lower values indicate a more complex structure, increasing the difficulty of spatial understanding. Based on the intelligibility value, spaces can be classified into three categories[48] (Table 6). ”

3.3.4. Isovist Area, Visual Integration and Extended Metrics

Specific content added: “Benedikt.[49] emphasizes that the isovist area refers to the area of the spatial region fully visible from a given observation point; the larger the isovist area, the broader the visual extent covered by the space and the greater its openness and visibility…….Huang and Lee.[28] examined the integration of Baidu heatmaps with space syntax and found a statistically significant correlation between integration and visitor clustering intensity.”

5.Comment:Ensure figures referenced (e.g., PRISMA diagram, Fig. 3) are embedded correctly in final layout.

Response: Thank you for this reminder. We have carefully checked all figures in the revised manuscript, to ensure that they are correctly referenced and embedded in the final layout.

6.Comment:Writing is mostly clear but would benefit from further proofreading for redundancy and flow.

Response: Thank you for this valuable comment. We have carefully proofread the entire manuscript to improve readability, reduce redundancy, and enhance the overall flow of the text. We believe these revisions have significantly improved the clarity and coherence of the paper.

Reviewer #2:

1.Comment:The manuscript examines a pertinent and timely subject with potential significance for both scholarly inquiry and practical implementation. The study design seems methodologically robust, and the data provided corroborate the overarching conclusions. The statistical analyses utilise suitable methods and seem to have been conducted with sufficient rigour, enhancing the reliability of the findings. The manuscript is predominantly well-composed, with arguments articulated in a coherent and logical fashion; however, minor grammatical revisions and occasional rephrasing could enhance its flow and readability. Certain terms and definitions require more precise clarification to enhance accessibility for readers from varied backgrounds.

Response: Thank you for this helpful comment. We have carefully proofread the manuscript and made minor grammatical revisions and wording adjustments to improve clarity, readability, and flow. Additionally, In the revised manuscript, we have provided clearer definitions of the technical terms used in the study. Specifically, these definitions have been added under Section 3.3 “Characteristics of space syntax indicators”, including:

3.3.1. Depth and Integration

Specific content added: “Mohamed et al.[30] emphasize that depth refers to the total number of steps required from a spatial unit to other spaces, which can reveal the marginal characteristics of the spatial structure; depth is effective for assessing less-frequented areas…… Conversely, lower integration values indicate that the space lies in peripheral and relatively enclosed areas, with a higher degree of enclosure and concealment[45].”

3.3.2. Control value and Choice

Specific content added: “Mohamed et al.[30] indicate that the control value measures the relative control of a spatial node over its adjacent nodes within a spatial network, reflecting the functional mechanism of the focal spatial unit in the local spatial structure and serving to assess its effects on surrounding spatial units…...Higher choice values indicate that the spatial unit plays a central role in route selection within the overall network[45].”

3.3.3. Connectivity and Intelligibility

Specific content added: “Wu et al.[31] state that connectivity refers to the number of links from a given spatial unit to its adjacent units, reflecting local connectedness…….Lower values indicate a more complex structure, increasing the difficulty of spatial understanding. Based on the intelligibility value, spaces can be classified into three categories[48] (Table 6). ”

3.3.4. Isovist Area, Visual Integration and Extended Metrics

Specific content added: “Benedikt.[49] emphasizes that the isovist area refers to the area of the spatial region fully visible from a given observation point; the larger the isovist area, the broader the visual extent covered by the space and the greater its openness and visibility…….Huang and Lee.[28] examined the integration of Baidu heatmaps with space syntax and found a statistically significant correlation between integration and visitor clustering intensity.”

2.Comment:Although the methodology is valid, the authors should offer more explicit justification for specific decisions regarding data handling, analysis, or variable selection to enhance the robustness of their approach. It would be advantageous to succinctly emphasise the originality and prospective impact of this research in the introduction, clearly articulating how it enhances current understanding. The manuscript possesses merit and, with minor enhancements in clarity and language refinement, can significantly contribute to the literature.

Response: Thank you for this constructive comment. In the revised manuscript, we have provided more explicit justification for the methodological framework. At the end of the Introduction, we added Fig. 1 “Flow diagram for literature review”, which visually illustrates the conceptual pathway and clarifies how data handling, analysis, and variable selection were systematically conducted. This addition strengthens the transparency and robustness of the methodology. Furthermore, we emphasized the originality and prospective impact of this research in the Introduction, Specific content added: “The necessity of this systematic review stems from the current paucity of comprehensive syntheses based on Space Syntax that analyze the interrelationship between garden spatial structure and tourist behavior. Although prior studies have confirmed the correlation between spatial configuration and tourist behavior, a lack of systematic reviews persists—particularly those integrating the explanatory power and adaptability of different spatial variables. ”

Reviewer #3:

1.Comment:This is an interesting article that accumulates recent developments in the use of spatial configuration metrics to determine patterns in tourist behaviour in gardens. The authors appear to have done a relatively comprehensive collection, final selection and description of research studies relying on existing models and standard processes. The manuscript is readable and intelligible though slightly repetitive. Where the article falters is in going beyond simple enumeration of the methods used in each study particularly as that relates to spatial configuration. While this is indeed an emerging domain with various published papers, the strategy for the review chosen here seems lacking. The authors have chosen to do a relatively shallow reading of the methodology of each article, focusing only on metrics, whereas research that is typically considered part of the space syntax domain has produced a multitude of combinations of spatial representations, algorithms, metrics and other methodological considerations such as radius (the spatial limit to the reach of the various algorithms). This approach would have worked fine if the sample was larger, but there are only 10 studies selected and all use rather different methods of getting to the result, not always equivalent. Because the sample is small, I would have expected a much deeper explanation of the spatial configuration methodologies. For example, a cursory reading of the studies suggests that they use different representations of space, but these are not mentioned in the manuscript. They do not appear as part of the methodology review or identified per-study. Some studies seem to use Visibility Graph Analysis (VGA) (referenced as 13, 15, 30 and 31), some Segment maps (28, 30, 32, 33) some Isovists (27, 31) and some Convex maps (20, 29). This can be problematic in some cases because metrics such as "Integration" and "Connectivity" produce different values for different representations at the same location (especially VGA and Segment maps) making comparison to human behaviour and perception difficult. Given the small sample I would have also at least expected the authors to further provide examples of the use of these methods in other scales of the built environment.

Response: Thank you for this thoughtful comment. We have strengthened the review strategy and deepened the methodological synthesis as follows:

a. Clarifying the review strategy — We added a visual overview of the research process at the end of the Introduction, presented as Fig. 1 “Flow diagram for literature review.” This figure makes the review strategy and steps explicit.

b. Updating the study sample and screening — In Section 2.3 “Study selection,” we re-screened the literature and updated the final sample to 18 studies, and we revised the PRISMA chart accordingly as Fig. 2 “PRISMA flow diagram.” This explicitly documents the sample size and screening pathway.

c. Deepening the methodological synthesis — In Section 3.1 “Research characteristics,” we added the following content and a new table to move beyond a simple enumeration of metrics:

Specific content added: “This study provides a systematic synthesis of the 18 included studies (Table 4), summarizing the following dimensions: study location; spatial representations (VGA, Segment, Isovist, Axial); space syntax metrics (e.g., Integration, Connectivity, Choice, Control, Depth, Intelligibility); radius/weight settings; and behavioral outcomes. Due to differences in computational foundations and scale semantics across representations, identically named metrics (Integration, Connectivity) may yield different values at the same location and lead to different interpretations, with the discrepancy between VGA and Segment being particularly pronounced. To ensure comparability, we group by representation during synthesis and interpret the relationships between metrics and behavior/perception under similar radius/weight settings, avoiding direct numerical comparisons across representations.”

We also added Table 4 “Characteristics of the studies.” which, for each study, records the spatial representation, metrics, radius/weight settings, study sites, and behavioral outcomes, thereby providing a transparent, per-study methodological profile rather than a metric-only listing.

d. Term definitions to support cross-representation reading — In Section 3.3 “Characteristics of space syntax indicators” (subsections 3.3.1–3.3.4), we added concise definitions and explanations for core indicators (Depth/Integration; Control/Choice; Connectivity/Intelligibility; Isovist Area/Visual Integration and extended

---

## [Decision Letter · Decision Letter 1]

30 Oct 2025

Dear Dr. Li,

Thank you for submitting your manuscript to PLOS ONE. After careful consideration, we feel that it has merit but does not fully meet PLOS ONE’s publication criteria as it currently stands. Therefore, we invite you to submit a revised version of the manuscript that addresses the points raised during the review process.

We look forward to receiving your revised manuscript.

Kind regards,

Yile Chen, Ph.D. in Architecture

Academic Editor

PLOS ONE

**Journal Requirements:**

**Additional Editor Comments:**

Currently, half of the reviewers have accepted the manuscript, but there are still some minor issues, such as formulas, that need to be corrected and verified. We hope the authors can revise these as soon as possible.

Reviewers' comments:

Reviewer's Responses to Questions

**Comments to the Author**

Reviewer #1: All comments have been addressed

Reviewer #2: All comments have been addressed

Reviewer #3: (No Response)

Reviewer #4: All comments have been addressed

2. Is the manuscript technically sound, and do the data support the conclusions?

Reviewer #1: Yes

Reviewer #2: Yes

Reviewer #3: Partly

Reviewer #4: Yes

3. Has the statistical analysis been performed appropriately and rigorously?

Reviewer #1: Yes

Reviewer #2: Yes

Reviewer #3: N/A

Reviewer #4: Yes

4. Have the authors made all data underlying the findings in their manuscript fully available?

Reviewer #1: Yes

Reviewer #2: Yes

Reviewer #3: Yes

Reviewer #4: Yes

5. Is the manuscript presented in an intelligible fashion and written in standard English?

Reviewer #1: Yes

Reviewer #2: Yes

Reviewer #3: Yes

Reviewer #4: Yes

**Reviewer #1: ** (No Response)

**Reviewer #2:**  This manuscript, "The Influence of Garden Spatial Configuration on Tourist Behaviour: A Systematic Review Based on Space Syntax," conducts a detailed review to investigate a pertinent and emerging subject that connects perception, spatial configuration, and tourist behaviour. The revision demonstrates a significant enhancement in the theoretical framing, clarity, and structure.

Advantages:

The study now adheres to PRISMA 2020 with rigour, utilising transparent inclusion criteria and quality appraisals (CCATs). The methodology and research questions are explicitly stated. The new Structure, Perception and Behaviour (SPB) framework and cross-scale comparison significantly improve the conceptual depth. Key Space Syntax metrics are defined, and tables and figures have been revised to enhance consistency and readability.

Suggestions for Minor Enhancement:

Clarify whether the SPB framework is a synthesis from the literature or a proposed theoretical model. Provide a concise explanation of the correlation between behavioural outcomes (e.g., cognitive ease, legibility) and metrics such as integration and choice. Include a brief commentary on the cultural distinctions between Western and Eastern garden typologies. The final round of English language and formatting refinement will be conducted.

Overall:

The manuscript is conceptually valuable and methodologically robust. It will make a substantial contribution to the literature on tourist behaviour and spatial configuration with some minor theoretical clarification and language polishing.

**Reviewer #3:**  Thank you for taking the time to respond and address the issues in the paper. Here are some more comments:

1. In response to Reviewer #3, 5. Comment Response: I mentioned Connectivity and Isovist Area as the same exactly _because both studies use VGA_ in which case they are the indeed practically the same. The original Hillier Connectivity refers to axial connections (essentially the number of junctions of a street), but in VGA Connectivity (or Visual Connectivity if you'd rather make this clearer) refers to the number of cells that are visible from a cell. This is calculated by creating a 360-degree isovist from each cell and looking at the number of cells that fall within the isovist polygon. The larger the area of the isovist polygon (what Isovist Area measures), the more the cells; and the smaller the cells, the more Visual Connectivity will approach Isovist Area. There is a finer point about intervisibility, but the reality of the VGA grid cell construct is that there are no non-reciprocal relationships (i.e. every cell that cell A can "see" can also "see" A). This is exactly because we use 360-degree isovists and because there are no studies (as far as I'm aware) that actually look at VGA with one-way visibility materials (though even in that case the practical - what the code outputs - result will once again be the same). If you'd like to dig deeper into the details of VGA metric calculation look at Koutsolampros (2021) section 4.3.1 on page 124, or Koutsolampros et al. (2019).

2. In section 3.1 it is not immediately clear that you mean that there are identically named metrics across representations. I would rephrase otherwise it seems that Integration and Connectivity are somehow identically named between them.

3. While table 4 is certainly welcome it seems that there has not been enough attention to detail:

3.a. For Huang & Lee (2023), "Kernel density estimation" and "heatmap" are not spatial representations.

3.b. For Chen & Yang (2025), what are "Isovist fields"? These are not explained anywhere.

3.c. For Mohammadi & Ujang (2021), what are experiential maps? Not explained anywhere. Are they spatial representations? How?

3.d. For Yu et al. (2021), it seems the authors use convex, not axial, even if they mention axial in some places.

3.e. For Zhang et al. (2019), you say "VGA + Depthmap", but Depthmap is just the tool for (among other things) carry out VGA.

3.f. For Chen et al. (2025), you say "Isovist polygons & Parameters", why not just "Isovists", why is this different to "Isovist fields" above, and what are "Parameters" (certainly not a representation).

4. In section 3.3 "visual" indicators are also "topological" due to the way that the grid is constructed (see comment 1 and related references). Either actually state the difference in measurement (topological/visual, metric, angular), or just actually specify that these refer to different representations (segment, convex, grid/VGA).

5. In 3.3.1 you are describing Topological/Visual or "Step" depth (number of steps to get somewhere). There's also metric depth and angular depth. Please specify the name (and ideally everywhere else).

6. In 3.3 generally; I don't understand why the grouping of metrics in this way (and it's not explained). Why "Control value and Choice"? Why does Control have "value" next to it and Choice does not? Why are Connectivity and Intelligibility in the same section? Please revise and group the metrics in a more meaningful way.

7. In 3.3.3, formula (6) doesn't mean anything. Intelligibility comes from using a specific statistical process - please explain it and specify the formula.

8. In 3.3.4, it is a strange statement that "visual integration" is good for lingering and photography given that areas with high integration tend to be intermediate central pathways (correctly assumed to be generating social interaction). If this is from the authors then specify the exact page otherwise explain.

9. In 3.3.4 (again), the "Common forms" mentioned are of combinations, not of metrics, making this harder to understand. What kind of metrics come out of the combinations with GPS trajectories? Also, questionnaire results are usually output or "dependent" metrics, so I wouldn't include them as metrics that may influence human behaviour or perception.

10. I noticed in 4.1 that there is a full stop after the names of authors even if they are just one or two (and not "et al."), for example with "In addition, Lee.[34] elucidated the relationship". Please correct throughout.

11. In section 6 "Different spatial representations" should have a space prior to the previous full stop.

**Reviewer #4:**  PLOS ONE, Manuscript Number: PONE-D-25-34363R1

Title: The Influence of Garden Spatial Configuration on Tourist Behavior: A Systematic

Review Based on Space Syntax

As a reviewer assessing the revised manuscript (Number: PONE-D-25-34363R1), I am satisfied with the revisions made by the authors. They have addressed all of my previous comments and suggestions comprehensively, which has strengthened the manuscript. The manuscript is now in substantially better shape, and I have no further comments at this stage.

**Do you want your identity to be public for this peer review?** For information about this choice, including consent withdrawal, please see our Privacy Policy

Reviewer #1: **Yes: ** Adib Amany

Reviewer #2: **Yes: ** SAYON PRAMANIK

Reviewer #3: No

Reviewer #4: **Yes: ** Shatha Malhis

---

## [Author Response · Author response to Decision Letter 2]

7 Dec 2025

Response to Reviewers' Comments

Title: The Influence of Garden Spatial Configuration on Tourist Behavior: A Systematic Review Based on Space Syntax

Manuscript Number : PONE-D-25-34363R1

We sincerely thank the academic editor and reviewers for their valuable feedback. We have addressed all comments and revised the manuscript accordingly. Our detailed responses are provided below.

Journal Requirements:

Response: Thank you for the guidance. In line with the journal’s requirement, we carefully reviewed the publications mentioned by the reviewer. Following an independent relevance check, we found that these works are not essential to the specific focus of our review (garden-scale spatial configuration and tourist behavior using space syntax) and would introduce overlap without adding substantive value. Accordingly, we have removed those citations in the revision.

Response: We have rechecked all references one by one to ensure completeness, accuracy, and retraction status. To the best of our knowledge at this revision stage, none of the cited works have been retracted. If any item is later found to be retracted, we will replace it with the most current and relevant source in accordance with the journal’s policy.

Reviewer #1: (No Response)

Reviewer #2:

1.Suggestions for Minor Enhancement:

(1) Clarify whether the SPB framework is a synthesis from the literature or a proposed theoretical model.

Response: Thank you for this helpful suggestion. We now explicitly clarify the provenance of the SPB framework in Section 4.3.Structure-Perception-Behavior (SPB) framework : it is a theory-informed synthesis from the literature, not a wholly new theory. Building on Mehrabian & Russell’s Stimulus–Organism–Response (SOR) model and Rapoport’s Culture–Environment–Behavior (CEB) framework, we articulate the mediating mechanism as X (spatial structure via space-syntax metrics) → M (tourist perception: aesthetics, legibility, cultural cognition, safety, restoration) → Y (behavioral outcomes: route choice, dwell time, social interaction), and construct the resulting Structure–Perception–Behavior (SPB) framework. The revised wording is: “Based on a systematic review of the relevant literature, and drawing on Mehrabian and Russell’s SOR model and Rapoport’s CEB explanatory framework, this study clarifies the “X → M → Y” mediating mechanism and constructs a “structure–perception–behavior” (SPB) framework.”

(2) Provide a concise explanation of the correlation between behavioural outcomes (e.g., cognitive ease, legibility) and metrics such as integration and choice.

Response: Thank you for the suggestion. We have added a concise explanation in Section 4.3 “Structure–Perception–Behavior (SPB) framework.” The new text clarifies how behavioural outcomes (e.g., cognitive ease and legibility) relate to key space-syntax metrics:

“Spaces with high integration typically exhibit greater visual centrality and spatial accessibility. Their clear orientation and easily recognizable routes enhance the spatial configuration's identifiability and legibility. Accordingly, the more recognizable and legible a space is, the higher its integration tends to be. In addition, high Choice increases the likelihood that a path functions as a primary route, and the formation of primary routes, in turn, reinforces high Choice. Through this mutual influence, high-choice main routes often become centers of human activity and aggregation and are more prone to high pedestrian flow and congestion. Consequently, Choice is positively associated with behavioral outcomes: the higher the Choice value, the greater the probability that a path will be traversed, thereby shaping tourists' route preference decisions.”

(3) Include a brief commentary on the cultural distinctions between Western and Eastern garden typologies.

Response: Thank you for the helpful suggestion. We have added a brief commentary on cultural distinctions between Western and Eastern garden typologies in Section 1. Introduction. The new paragraph reads:

“Across different civilizational lineages, the evolutionary trajectories and spatial connotations of garden types differ markedly. For instance, Eastern gardens, influenced by Confucian, Taoist, and Buddhist philosophies, emphasize the creation of spaces that embody the "unity of heaven and humanity," personal cultivation, and transcendent mental states. Within this framework, Chinese imperial gardens, shaped by ritual systems and imperial discourse, emphasize grand layouts and central axis order, symbolizing political power. Private gardens, however, favored the "microcosmic" landscape aesthetic, emphasizing the literati's appreciation of shifting vistas and self-cultivation through "scenes that change with every step."Japanese gardens, inheriting early Chinese Buddhist traditions, ultimately evolved under the influence of Zen and the tea ceremony to use stones, sand, and moss as primary elements, creating wabi-sabi aesthetics and meditative spaces. In contrast to the Eastern pursuit of natural beauty, Western gardens emphasize the unity of religion and power through converging axes and waterways. Examples include Renaissance gardens that express "rational domination over nature" through geometric order, and Baroque gardens that reinforce monarchical authority through spatial hierarchy. Within contemporary urban contexts, gardens, whether rooted in Eastern traditions or Western lineages, have become spatial vessels for recreation, sightseeing, and social interaction for both residents and visitors. ”

(4) The final round of English language and formatting refinement will be conducted.

Response: We conducted in-depth revisions to key sections—the Abstract, Introduction, Conclusions, and Limitations—and applied targeted language and formatting edits to the remaining content to improve clarity and coherence. No substantive changes were made to the study design, data, or results.

Reviewer #3:

1.In response to Reviewer #3, 5. Comment Response: I mentioned Connectivity and Isovist Area as the same exactly _because both studies use VGA_ in which case they are the indeed practically the same. The original Hillier Connectivity refers to axial connections (essentially the number of junctions of a street), but in VGA Connectivity (or Visual Connectivity if you'd rather make this clearer) refers to the number of cells that are visible from a cell. This is calculated by creating a 360-degree isovist from each cell and looking at the number of cells that fall within the isovist polygon. The larger the area of the isovist polygon (what Isovist Area measures), the more the cells; and the smaller the cells, the more Visual Connectivity will approach Isovist Area. There is a finer point about intervisibility, but the reality of the VGA grid cell construct is that there are no non-reciprocal relationships (i.e. every cell that cell A can "see" can also "see" A). This is exactly because we use 360-degree isovists and because there are no studies (as far as I'm aware) that actually look at VGA with one-way visibility materials (though even in that case the practical - what the code outputs - result will once again be the same). If you'd like to dig deeper into the details of VGA metric calculation look at Koutsolampros (2021) section 4.3.1 on page 124, or Koutsolampros et al. (2019).

Response: Thank you for the clear explanation and the references. We sincerely appreciate your professional guidance—this suggestion has significantly improved the precision of the manuscript’s terminology.

2. In section 3.1 it is not immediately clear that you mean that there are identically named metrics across representations. I would rephrase otherwise it seems that Integration and Connectivity are somehow identically named between them.

Response: Thank you for flagging the potential ambiguity. To avoid the impression that metrics are identically named across different representations, we have revised Section 3.1 (“Research characteristics”). The paragraph now reads:

“This study provides a systematic synthesis of the 18 included studies (Table 4). The general characteristics were summarized across the following dimensions: country of publication, study location, spatial analysis, and space syntax modeling methods, core space syntax metrics (integration, connectivity, choice, control), radius or weighting settings, and reported outcomes. Specifically, spatial analysis and space syntax modeling methods comprised two categories: (1) visualization- or statistics-based analyses, including kernel density estimation (KDE), heatmaps, and experiential maps; and (2) space syntax modeling methods, including axial maps, visibility graph analysis (VGA), segment analysis, and isovist-based models. Notably, axial, segment, and VGA analyses emphasize the global network structure or connectivity, whereas the isovist analysis focuses on local field-of-view visibility”.

3. While table 4 is certainly welcome it seems that there has not been enough attention to detail:

3.a. For Huang & Lee (2023), "Kernel density estimation" and "heatmap" are not spatial representations.

Response: Thank you for the reviewer’s correction. We agree that Kernel Density Estimation (KDE) and heatmaps are not spatial representation/modeling methods. We have corrected Table 4: Characteristics of the studies, reclassifying them under “Visualization and statistical analysis” rather than “Spatial representation/model,” and we have updated the table headers and footnotes accordingly and standardized the terminology throughout the manuscript.

In addition, for reasons of scholarly transparency and copyright compliance: most illustrative figures are owned by the original articles or institutions and should not be republished without explicit permission; even with attribution, such reuse may exceed fair-use limits. Therefore, we have made minor structural adjustments to Table 4. Characteristics of the studies and, in the table notes, direct readers to consult the original figures via the corresponding reference numbers.

3.b. For Chen & Yang (2025), what are "Isovist fields"? These are not explained anywhere.

Response: Thank you for pointing this out. Upon verification, we found that Chen & Yang (2025) should in fact be Chen & Yang (2023). This has been corrected in Table 4. Characteristics of the studies, and the modeling method has been explicitly identified as “Isovist (visibility model).” To avoid ambiguity, we have replaced “isovist fields” with “isovist” throughout the manuscript and added the following note in the main text and in the Table 4 notes: Isovist: the portion of space visible from a given observation point under conditions of spatial occlusion.

3.c. For Mohammadi & Ujang (2021), what are experiential maps? Not explained anywhere. Are they spatial representations? How?

Response: Thank you for the clarification request. We have clarified in the table notes that “experiential maps” are defined as: “Experiential maps: records of participants’ subjective experiences and behaviors within the site.” Accordingly, in Table 4. Characteristics of the studies, we have updated the column header to “Spatial Analysis and Modeling Methods” to avoid ambiguity.

3.d. For Yu et al. (2021), it seems the authors use convex, not axial, even if they mention axial in some places.

Response: Thank you for the helpful observation. We have corrected the entry for Yu et al. (2021) in Table 4. Characteristics of the studies: the space-syntax representation has been updated from “axial” to “convex.”

3.e. For Zhang et al. (2019), you say "VGA + Depthmap", but Depthmap is just the tool for (among other things) carry out VGA.

Response: Thank you for the correction. To improve accuracy, we have updated Table 4. Characteristics of the studies by changing Zhang et al. (2019) from “VGA + Depthmap” to VGA only.

3.f. For Chen et al. (2025), you say "Isovist polygons & Parameters", why not just "Isovists", why is this different to "Isovist fields" above, and what are "Parameters" (certainly not a representation).

Response: Thank you for the suggestion. We have standardized the terminology in Table 4: Characteristics of the studies, correcting “Isovist polygons & Parameters” to “Isovists ”, and added a footnote definition to avoid ambiguity. Specifically: “Isovist: the visible space from a given point under conditions of spatial occlusion.”

4. In section 3.3 "visual" indicators are also "topological" due to the way that the grid is constructed (see comment 1 and related references). Either actually state the difference in measurement (topological/visual, metric, angular), or just actually specify that these refer to different representations (segment, convex, grid/VGA).

Response: Thank you for the constructive comment. We have rewritten Section 3.3, “Characteristics of Space Syntax Indicators,” to eliminate ambiguity and improve readability. The specific revisions are as follows:

“Space Syntax is a theoretical and methodological framework for analyzing the relationship between spatial structures and human behavior [46]. Spatial configuration metrics derived from Space Syntax modeling, such as those based on axial maps and segment maps, primarily include integration, connectivity, depth, control, mean depth, and relative asymmetry[47], these metrics focus on spatial accessibility and connectivity. Visual features mainly include visual connectivity, visual integration, visual selectivity, visible area, and visible boundary length[48], these are based on the VGA (Visual Area Gauge) in spatial syntactic modeling methods, and these metrics focus on visual accessibility and visual perception. Therefore, both theoretically possess topological properties, but there are certain differences in their spatial syntactic modeling methods. The 18 articles included in this study cover topics such as depth value, integration, control value, selectivity, connectivity, visible area, visual integration, and composite indicators. To further understand the core indicators of Space Syntax, this study integrates spatial attributes and their metric characteristics to provide a detailed elaboration on spatial structure and accessibility, local control and path flow, local connectivity, spatial intelligibility, isovist area and visual integration, as well as extended metrics.”

5. In 3.3.1 you are describing Topological/Visual or "Step" depth (number of steps to get somewhere). There's also metric depth and angular depth. Please specify the name (and ideally everywhere else).

Response: Thank you for the helpful suggestion. We have revised Section 3.3.1 “Core Space Syntax Metrics”, The specific revisions are as follows:

“Within the framework of Space Syntax theory, depth is used to analyze the topological, metric, and angular relationships between spatial units, and it is generally categorized into three types: Step Depth, Metric Depth, and Angular Depth. Step Depth is calculated based on the number of steps along a path and reflects the hierarchical relationships within the overall structure of spatial units; Metric Depth is computed using the geometric length of space and emphasizes the actual physical distance and walking cost in space; Angular Depth is calculated mainly based on changes in turning angles along the path and emphasizes peopl

---

## [Editor Report · Decision Letter 2]

15 Dec 2025

The Influence of Garden Spatial Configuration on Tourist Behavior: A Systematic Review Based on Space Syntax

PONE-D-25-34363R2

Dear Dr. Li,

We’re pleased to inform you that your manuscript has been judged scientifically suitable for publication and will be formally accepted for publication once it meets all outstanding technical requirements.

Kind regards,

Yile Chen, Ph.D. in Architecture

Academic Editor

PLOS One
---

## [Editor Report · Acceptance letter]

PONE-D-25-34363R2

PLOS One

Dear Dr. Li,

I'm pleased to inform you that your manuscript has been deemed suitable for publication in PLOS One. Congratulations! Your manuscript is now being handed over to our production team.

Kind regards,

on behalf of

Dr. Yile Chen

Academic Editor

PLOS One